

1          A New Instrument for Time Resolved Measurement of HO$_2$ Radicals

2          Thomas H. Speak[1], Mark A. Blitz[1,2]*, Daniel Stone[1] and Paul W. Seakins[1,2]*

3          *1 – School of Chemistry, University of Leeds, Leeds, LS2 9JT, UK*

4          *2 – National Centre for Atmospheric Science, Leeds, LS2 9JT, UK*

*Abstract*
OH and HO$_2$ radicals are closely coupled in the atmospheric oxidation and combustion of volatile
organic compounds (VOCs). Simultaneous measurement of HO$_2$ yields and OH kinetics can
provide the ability to assign site specific rate coefficients that are important for understanding the
oxidation mechanisms of VOCs. By coupling a FAGE LIF detection system for OH and HO$_2$ with
a high pressure laser flash photolysis system, it is possible to accurately measure OH pseudo-first-
order loss processes up to ~100000 s$^{-1}$ and to determine HO$_2$ yields via time resolved
measurements. This time resolution allows discrimination between primary HO$_2$ from the target
reaction and secondary production from side reactions. The apparatus was characterized by
measuring yields from the reactions of OH with H$_2$O$_2$ (1:1 link between OH and HO$_2$), with
C$_2$H$_4$/O$_2$ (where secondary chemistry can generate HO$_2$), with C$_2$H$_6$/O$_2$ (where there should be
zero HO$_2$ yield) and with CH$_3$OH/O$_2$ (where there is a well-defined HO$_2$ yield).
As an application of the new instrument, the reaction of OH with n-butanol has been studied at
293 and 616 K. The bimolecular rate coefficient at 293 K, $(9.24 \pm 0.21) \times 10^{-12}$ cm$^3$ molecule$^{-1}$ s$^{-1}$,
is in good agreement with recent literature, verifying that this instrument can both measure HO$_2$
yields and accurate OH kinetics. At 616 K the regeneration of OH in the absence of O$_2$, from the
decomposition of the β-hydroxy radical, was observed, which allowed the determination of the
fraction of OH reacting at the β site $(0.23 \pm 0.04)$. Direct observation of the HO$_2$ product in the
presence of oxygen has allowed the assignment of the α-branching fractions $(0.57 \pm 0.06)$ at 293 K
and $(0.54 \pm 0.04)$ at 616 K; branching ratios are key to modelling the ignition delay times of this
potential 'drop-in' biofuel.

## 27   1 Introduction

In the atmosphere, HO$_2$ and OH radicals (OH + HO$_2$ = HOx) are closely coupled via several
reactions as shown in Scheme 1. The short lifetimes of HOx radicals mean that concentrations are
determined by chemical production and removal and not by transport processes, making them ideal
candidates as test species for our understanding of atmospheric chemical mechanisms (Stone et
al., 2012;Monks, 2005;Stockwell et al., 2012).




**Scheme 1.** A simplified tropospheric HOx cycle showing the importance of these short-lived radical species both to the chemical removal of VOCs and the formation of ozone.


In Scheme 1, the reaction of alkoxy radicals with molecular oxygen is a major route to $HO_2$ formation; however, this is not the only significant $HO_2$ formation process; for example, in the atmospheric oxidation of n-butanol, $HO_2$ can be formed via two different mechanisms. Abstraction by OH at the α position leads to a radical which reacts with oxygen to directly produce $HO_2$ (R1a, R2) whereas abstraction at other sites leads to alkylperoxy radical ($C_4H_9O_2$) formation with varying fractions of the $RO_2$ forming alkoxy radicals, and subsequently $HO_2$ (McGillen et al., 2013) on a longer timescale.

$$OH + CH_3CH_2CH_2CH_2OH \rightarrow CH_3CH_2CH_2CHOH + H_2O \tag{R1a}$$

$$CH_3CH_2CH_2CHOH + O_2 \rightarrow HO_2 + CH_3CH_2CH_2CHO \tag{R2}$$

The fraction of alkoxy radicals formed depends on the mechanism of $RO_2$ removal (reaction with NO or self or cross-reactions) and the yield of $HO_2$ from the alkoxy radical depends on the competition between decomposition, isomerization and reaction with $O_2$, which in turn will depend on the structure of the alkoxy radical, temperature, pressure and concentration of oxygen (Orlando et al., 2003). Therefore, in order to determine the $HO_2$ yield from the OH initiated oxidation of compounds such as n-butanol, it is important to have a selective, sensitive and time resolved method of $HO_2$ detection.

The importance of $HO_2$ chemistry is not limited to atmospheric processes; $HO_2$ is a key intermediate in low temperature (500 – 1000 K) combustion processes, particularly those involving



oxygenated fuels (Zador et al., 2011). The mechanisms of low temperature combustion are of
particular interest in the development of new engine technologies such as reactively controlled
compression ignition (RCCI) (Reitz and Duraisamy, 2015) and are closely linked to atmospheric
oxidation mechanisms. Monitoring $HO_2$ concentrations under the elevated temperatures and high
pressures of combustion processes is therefore also of interest. In low-temperature combustion,
$HO_2$ formation is a chain inhibition process, with OH reformation a chain propagating or chain
branching process. The ratio of chain branching to chain inhibition processes is often the
controlling factor in modelling ignition delay times (Agbro et al., 2017). High temperatures and
concentrations of oxygen may be required to convert atmospheric processes, which take several
10s of seconds at ambient temperatures (and hence may be influenced by surface chemistry or
secondary reactions) to the milli- or microsecond timescale where they can be studied by flash
photolysis techniques without such interferences (Medeiros et al., 2018).
Direct measurements of $HO_2$ rely on absorption techniques, and kinetic information on
$HO_2$ reactions has been determined mainly using absorption spectroscopy. This can be achieved
either with conventional absorption techniques, often in the UV, (including multipass optics to
enhance the pathlength) or in the IR with cavity ring down spectroscopy (CRDS) (Assaf et al.,
2018;Onel et al., 2017). However, the $HO_2$ UV absorption spectrum (200 - 260 nm) is broad and
featureless (Crowley et al., 1991), and as such, overlaps with the UV absorptions of many other
species present in atmospheric degradation pathways or combustion systems (particularly $H_2O_2$
and $RO_2$). To utilize the selectivity of the structured IR spectra, absorption methods have been
developed in both the mid and near-IR (NIR) (Taatjes and Oh, 1997). Mid-IR absorption features
for $HO_2$ provide sufficient absorption cross-sections for study (Jemialade and Thrush, 1990) but
suffer from severe pressure broadening, reducing sensitivity under the conditions relevant to
atmospheric and combustion systems (Thiebaud and Fittschen, 2006). Detection in the NIR has
similar advantages in terms of a structured spectrum providing greater selectivity; the weaker
absorption cross-sections are compensated by the higher powers and ease of use of NIR laser
sources (Gianella et al., 2016). However, pressure broadening and interference from $H_2O$
absorptions can make these measurements difficult at even low concentrations of water ($10^{14}$
molecule cm$^{-3}$).





In the atmosphere (Stone et al., 2012) and in chamber studies (Glowacki et al., 2007), $HO_2$
is detected using a sensitive, but indirect method via conversion to OH, with detection of OH via
laser induced fluorescence (LIF) (Hard et al., 1984;Brune et al., 1995;Fuchs et al., 2011) or
conversion to $H_2^{34}SO_4$ with subsequent detection of the acid via mass spectrometry (Edwards et
al., 2003;Hanke et al., 2002). In the LIF method, also known as Fluorescence Assay by Gaseous
Expansion (FAGE (Hard et al., 1984)), which is the technique used in this study, OH is sampled
into a low pressure region through a pinhole. Low pressures allow for the temporal separation of
resonant 308 nm fluorescence from the excitation pulse. Following the first detection axis for OH,
a flow of NO is introduced which reacts with $HO_2$ (R3):
$HO_2 + NO \rightarrow OH + NO_2$                                                                    (R3)
The resulting OH is monitored at a second detector. The high sensitivity with which OH can be
detected gives $HO_2$ detection limits in the $10^8$ molecule cm$^{-3}$ range for $5-10$ s averaging, however,
to extract concentrations, both OH detection methods require calibration (Winiberg et al., 2015).
For chamber measurements of $HO_2$, comparisons with direct CRDS measurements have verified
the reliability of the calibration process (Onel et al., 2017).
$HO_2$ detection by LIF can be potentially sensitive to interferences from certain $RO_2$ species
which may also be converted to OH on short timescales. Interferences can be minimized by short
conversion times between NO injection and OH monitoring, utilizing low pressures, high flow
rates of the sample gas, and low NO concentrations to separate OH generation from $HO_2$ and $RO_2$,
reduced conversion of $HO_2$ reduces the sensitivity of this technique and as such in practice a
compromise between selectivity and sensitivity is used (Fuchs et al., 2011;Hard et al.,
1984;Whalley et al., 2013).
The current paper describes a significant development on our earlier FAGE based
instrument for time-resolved OH detection (Stone et al., 2016). In this improved system, laser flash
photolysis in a high pressure (up to 5 bar), temperature controllable ($300-800$ K) reactor (shown
in Figure 1) generated radicals which were then sampled through a pinhole forming a jet within
the low pressure detection region (shown in more detail in Figure 2). OH radicals were monitored
by LIF close to the pinhole. The jet breaks down after ~20 mm and NO was injected after this
point to convert some $HO_2$ into OH which was then detected by a second monitoring system. In
general, LIF becomes less sensitive at higher temperatures (due to distribution of population over



more rotational levels) and $O_2$ concentrations (due to quenching). Sampling into the low-pressure
region reduces both the effect of collisional quenching and temperature on the sensitivity of LIF
detection, although there is a reduction in the number density of the HOx species in the expansion.
We report the adaptation of our time-resolved OH-FAGE instrument to allow $HO_2$ detection, the
characterization of the instrument (including development of a calibration method for $HO_2$ yields
of OH initiated reactions), and the investigation of the influence of $RO_2$ species. Finally, we discuss
the application of the technique to determine the yield of $HO_2$ from the reaction of OH with n-
butanol.

### 123   2 Experimental

Reactions were carried out in a high pressure ($0.5 - 5$ bar) reaction cell which is described in
greater detail in Stone et al. (2016) and schematics of which are shown in Figures 1 and 2. The
high-pressure reactor was a 0.5 m stainless steel tube with a 22 mm internal diameter. Gas flows
were delivered to the high-pressure cell from a mixing manifold where calibrated mass flow
controllers (MFC) allowed for accurate control of flow rates. Low vapour pressure compounds:
OH precursors ($H_2O_2$), and substrates methanol and butanol, were delivered to the mixing manifold
from, thermostatted bubblers in pressure regulated backing flows of nitrogen ($N_2$). Ethane and
oxygen were delivered directly from cylinders into the mixing manifold through MFCs. The gas
flowrate through the cell was kept under laminar conditions with typical Reynolds values (Re) of
480 (corresponding to a flow rate for an experiment of 10 SLM at 2 bar); in general conditions
were maintained between 400-800 Re   (Re < 2400 = laminar flow), with some experiments carried
out with higher flowrates, up to 1800 Re.

Temperature control of the reactor between room temperature and 800 K was achieved by

altering the voltage applied to a coil heater (WATROD tubular heater, Watlow) over the last 30
cm of the stainless-steel tube. The heated region was fitted with a quartz liner (inner diameter 18
mm) to reduce wall-initiated chemistry. A temperature readout, from a type K thermocouple in the
gas flow, close to the pinhole, was calibrated for given flow rates, pressures and voltage settings
by measuring the highly temperature sensitive OH and methane rate coefficient, using the
temperature dependence reported by (Dunlop and Tully, 1993).  A more detailed description of
this method is described within instrument characterization (Section 3.4).





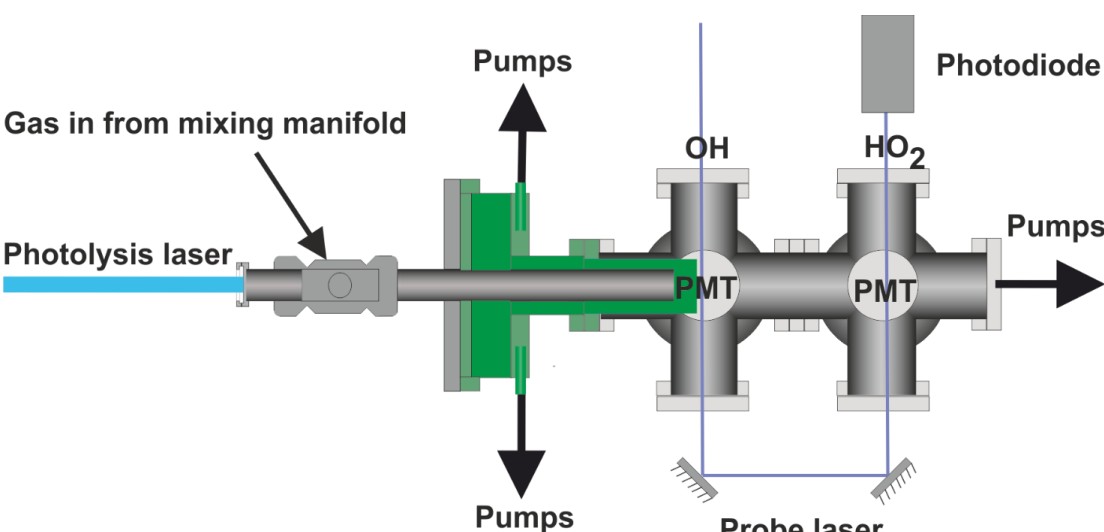


**Figure 1.** Schematic plan of the apparatus.


The photolysis of the OH precursor, $H_2O_2$, at 248 nm (Lambda Physik, Compex 200
operated using KrF) or 266 nm (frequency quadrupled Nd-YAG output, Quantel, Q-smart 850)
initiated the chemistry.
$H_2O_2 + h\nu \rightarrow 2OH$                                                                                         (R4)
Hydrogen peroxide was used as the OH precursor for all experiments where $HO_2$ detection was
performed, because it also acts as an internal calibrant to relate OH and $HO_2$, via reaction R5:
**OH** $+ H_2O_2 \rightarrow H_2O +$ **HO₂**                                                                 (R5)

However, in general, other OH precursors can be used. The OH precursor was maintained at low
concentrations ($1 \times 10^{14} - 1 \times 10^{15}$ molecule cm$^{-3}$) to minimise errors associated with assigning
pseudo-first-order kinetics for the loss of OH, and to reduce radical-radical reactions. Maintaining
a low radical precursor concentration had the additional advantage of minimising attenuation of
the photolysis beam, ensuring consistency in the initial radical concentrations generated along the
length of the high-pressure cell. Initial OH concentrations were in the range $2 \times 10^{11} - 5 \times 10^{13}$
molecule cm$^{-3}$.





A pinhole (diameter < 0.15 mm) at the end of the high-pressure reactor couples the reactor
to the low-pressure (0.3 – 5 Torr) detection cell. Details on OH detection can be found in Stone et
al. (2016). The accuracy of the instrument for OH measurement has recently been verified by
measurements of the rate coefficient of the reaction of OH with isoprene (Medeiros et al., 2018)
which are in excellent agreement with the literature. A more detailed schematic for the low-
pressure detection cell is shown in Figure 2.

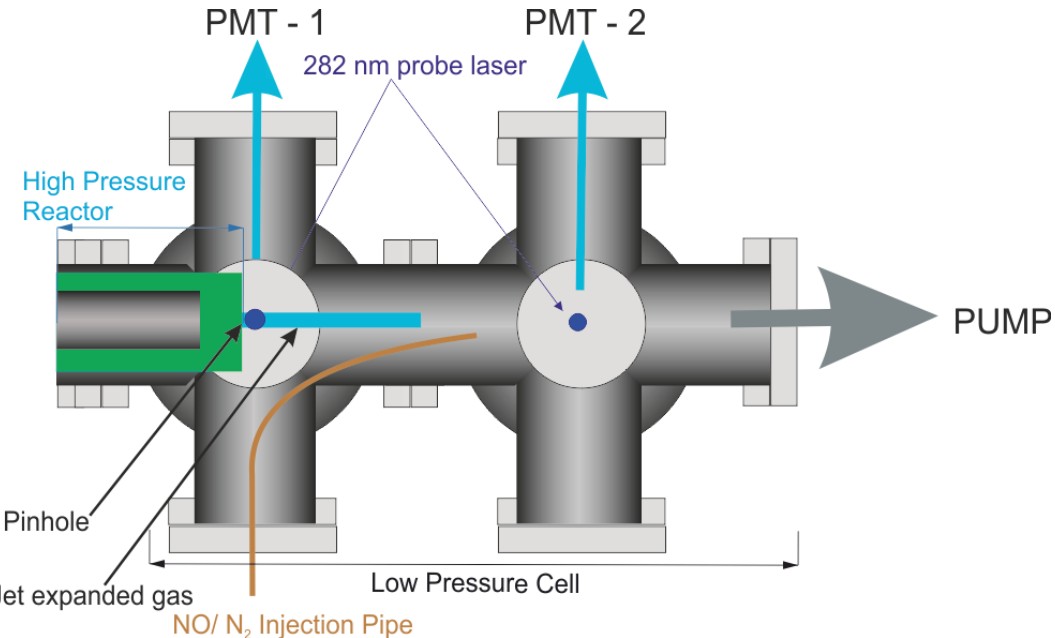

**Figure 2.** Detailed schematic elevation of the low-pressure detection region of the reactor. The
blue line represents the jet expanded gas; the jet breaks down after approximately 2 cm. NO was
injected through a 1.5 mm id stainless steel tube after the breakdown of the jet.

In the first low pressure detection cell, the OH was probed within the jet expanded gas,
close to pinhole (<5 mm), perpendicular to the gas flow. The OH was detected by off-resonance
laser induced fluorescence (LIF) at 308 nm following excitation with 282 nm light ($A^2\Sigma$ ($v' = 1$)
← $X^2\Pi(v'' = 0)$,$Q_{11}$). The 282 nm light was the frequency doubled output of a dye laser
((Rhodamine 6 G, Spectron) pumped at 532 nm by a Nd:YAG laser (Spectron), or (Rhodamine 6
G, Continuum) pumped by a Nd-YAG laser (Quantel, Q-smart 850)). Measuring the off-resonance





fluorescence allowed the use of a filter (308 ± 5 nm, Barr Associates) before the photomultiplier
(Perkin-Elmer C1943P) to remove scattered light and improved the signal to noise ratio.

A delay generator (BNC DG535) was used to vary the delay between the photolysis and

probe laser, facilitating generation of time profiles of the OH concentration. The traces were
scanned through multiple times (5 – 20) and the signal at each time point was averaged, giving
high precision OH loss traces. An example OH trace from the first detection cell for reaction R5
is presented in Figure 3. As reactions were carried out under pseudo first order conditions ([OH]
<< [substrate]), the time dependence of the OH LIF signal, $I_f$, (proportional to the [OH]) was given
by:

$$I_{f,t} = I_{f,0} e^{-k_{OH}t}$$

where $k_{OH} = k_5[H_2O_2]$. In Figure 3 two traces are presented from the first, OH, detection axis, these
two traces were taken in consecutive experiments with a constant $[H_2O_2]$ where the first trace
($k_{OH,1st} = (2351 ± 22)$ s$^{-1}$) was taken where $N_2$ was flowed into the low pressure region, the second
trace ($k_{OH,1st} = (2389 ± 18)$ s$^{-1}$) was taken when this flow had been switched to NO to allow $HO_2$
detection in the second detection cell, errors are given as 2 σ. The similarity of the OH decay traces
when either $N_2$ or NO was injected shows that there was no back streaming of NO in the low-
pressure cell and hence no $HO_2$ conversion at the first detection axis.

$HO_2$ radicals were monitored by the chemical transformation of $HO_2$ to OH via reaction

with NO (R3) in the low-pressure cell. Following the breakdown of the jet, after the Mach disk
(>2 cm beyond the pinhole), a small flow (5 sccm) of NO or $N_2$ was introduced  into the low-
pressure cell via a 1.5 mm i.d. stainless steel pipe (for a typical 0.5 Torr pressure in the FAGE cell
the NO concentration was $5.5 \times 10^{13}$ molecule cm$^{-3}$). After passing through the first detection cell,
the probe beam was redirected through the second low-pressure detection cell downstream of the
NO pipe allowing for the measurement of the OH concentration by LIF in the same manner as in
the first cell.



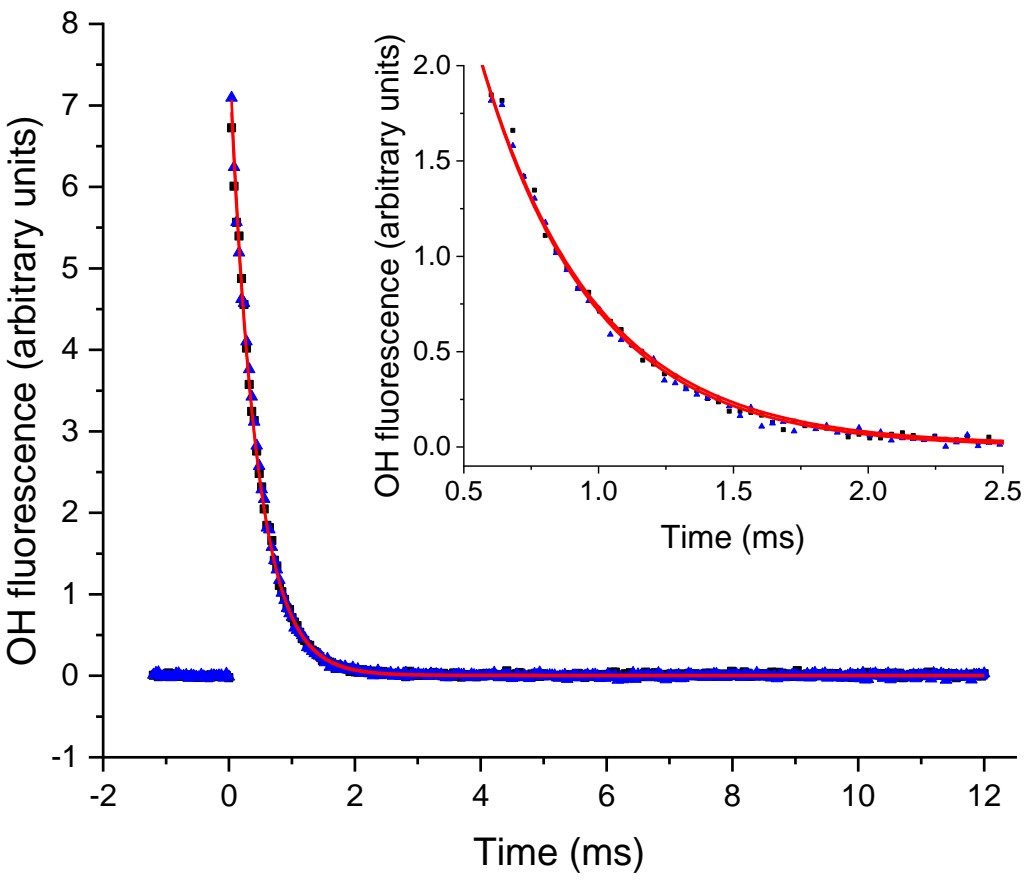


**Figure 3.** An example of the OH signal (blue triangles) collected at the first detection axis for the
reaction of OH with $H_2O_2$ ([$H_2O_2$] $\approx 1.4 \times 10^{15}$ molecule cm$^{-3}$, with a flow of $N_2$ into the low-
pressure cell, with black squares representing the subsequent trace taken with a flow of NO. The
red lines represent the non-linear least squares fits to an exponential decay ($k_{OH,1st} = (2351 \pm 22)$ s$^{-1}$
and $k_{OH,1st} = (2389 \pm 18)$ s$^{-1}$), 2 σ errors.


By switching between a flow of $N_2$ and NO, through the pipe, traces for OH loss and $HO_2$
formation could be elucidated, examples of which are shown in Figure 4. Subtraction of the two
OH traces in Figure 4, (upper, red trace is with NO injection and the signal corresponds to reactant
OH and OH produced from the titration of $HO_2$ to OH, lower, blue trace with $N_2$ injection is
reactant OH only) gave a resultant signal associated with $HO_2$ production in the high-pressure
reactor, shown the pink trace in Figure 4. The signal from the first PMT allowed for correction of



the signal heights at the second PMT for changes in the probe laser power or wavelength, any
variations in laser power or wavelength affect the absolute signal retrieved from both PMTs;
however, the relative signals retrieved from the PMTs remain consistent.

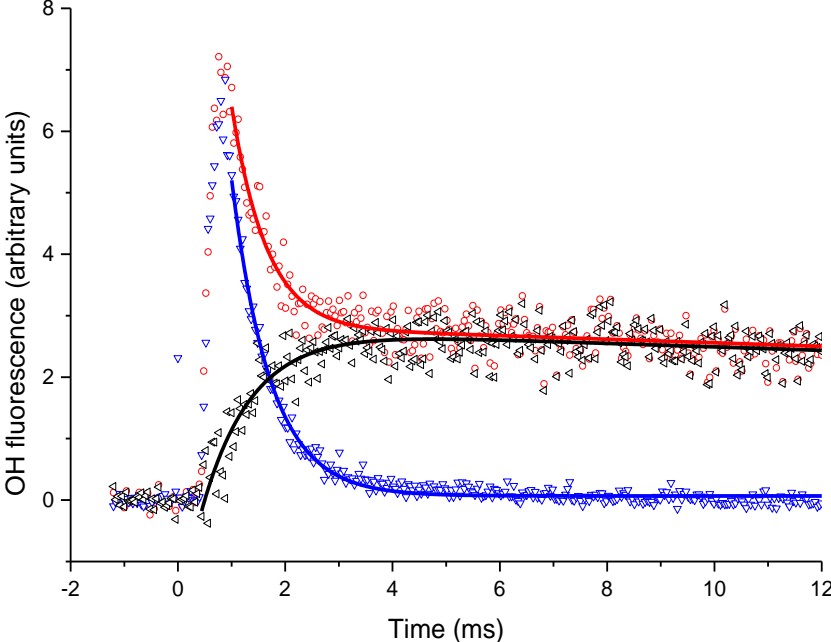


**Figure 4.** Examples of OH fluorescence traces collected at the second detection axis under the
same conditions for Figure 3. The blue triangles are where $N_2$ has been injected through the pipe,
i.e. no $HO_2$ to OH conversion. The OH signal profile differs from that in Figure 3, with $k_{OH,2nd} =$
$(1390 \pm 44)$ s$^{-1}$ (2 σ errors), additionally, there is a time-delay to peak OH, representing the
transport time (primarily the time taken to travel from the breakdown of the jet to the second
detection axis). The red circles are the OH signal obtained with NO injection. At short times the
signal is dominated by reactant OH, but at times greater than 2 ms, the signal is dominated by OH
titrated from the $HO_2$ product. The resultant OH trace associated with $HO_2$ formation in the high-
pressure cell  obtained by subtracting the two OH traces, obtained with either NO or $N_2$ injection
prior to the second detection axis, shown as black triangles, a biexponential growth and decay fit,
black curve, gave a formation rate coefficient, $k_{HO2,2nd} = (1080 \pm 150)$ s$^{-1}$ (2 σ error).

Fits to the $HO_2$ formation traces and OH loss traces from the second cell generated kinetic

parameters which differed from the accurate parameters collected at the first detection axis, $k_{OH,2nd}$



$= (1390 \pm 44)$ s$^{-1}$ and $k_{HO2,2nd} = (1080 \pm 150)$ s$^{-1}$ where the accurate loss parameters from the first
cell were $k_{OH,1st} = (2389 \pm 18)$ s$^{-1}$, 2 σ errors. This difference was the result of transport effects. By
comparison of the loss and formation parameters derived for OH + H$_2$O$_2$, for the first and second
detection cells, HO$_2$ formation rates could be assigned from a calibration plot (Figure 7).
Neither of the OH determinations in the two detection axes provide absolute measurements
of radical concentrations. Each detection axis could be calibrated as for chamber measurements,
but for our purposes a calibration reaction linking photolytically produced OH and HO$_2$ removes
many sources of error compared to an absolute calibration. The reaction of OH with the radical
precursor H$_2$O$_2$ which directly forms HO$_2$ with a 100 percent yield was used for calibration.
**OH** + H$_2$O$_2$ → H$_2$O + **HO$_2$**                                                                              (R5)
For reactions carried out where a reagent was added in addition to the H$_2$O$_2$, the resulting ratios
can be compared with those from the calibration reaction to allow assignment of an observed HO$_2$
yield. To assign the HO$_2$ yield from the test reaction required accounting for secondary HO$_2$
production in the high-pressure reactor, from OH + H$_2$O$_2$ and photolysis processes. From the
known rate coefficients, it was possible to calculate the fraction of OH reacting with the H$_2$O$_2$ and
hence the expected contribution to the HO$_2$ signal. Photolytic production of HO$_2$ was accounted
for by measuring the observed HO$_2$ signal in the absence of any H$_2$O$_2$.
In a typical experiment, the reaction of OH and H$_2$O$_2$ would be carried out four times, twice
in the absence of NO and twice with the addition of NO to calibrate the instrument. Exponential
fits to the OH decay as monitored in the first cell determine the peak OH signal. The OH signals
at the second detector recorded with only N$_2$ addition (reagent OH reaching the second detector)
and subtracted from the signal with NO added (reagent OH and HO$_2$) to give the net HO$_2$ signal.
This profile was fitted to, a biexponential growth and decay function, to extract the peak HO$_2$
signal for that set of conditions. Combinations of traces were then used to obtain an averaged value
(and uncertainty) of the signal on the first PMT (OH) to the net HO$_2$ signal at the second PMT for
this calibration reaction where OH reactant and HO$_2$ product have a 1:1 relationship. The same
process was then performed in the presence of the compound of study. The removal pseudo-first-
order rate coefficient with H$_2$O$_2$ and the reagent of study ($k'_{OH,1st} = k_{OH+H2O2}[H_2O_2]$ +
$k_{OH+TEST}[TEST]$) was compared to the removal pseudo-first-order rate coefficient with only H$_2$O$_2$





($k'_{OH,1st} = k_{OH+H2O2}[H_2O_2]$) to assign what fraction of the OH reacted with the $H_2O_2$ precursor and
hence the resulting contribution to the observed $HO_2$. Comparison of the remaining peak ratio to
the ratio from the $H_2O_2$ and OH calibration experiment provided the experimentally derived $HO_2$
yield for reaction of OH and the reagent of study.
Branching ratios to direct $HO_2$ formation could be assigned with an accuracy of ~10 %, the
limitations to this were signal to noise effects, where improved signal to noise could be achieved
by increasing the precursor concentration and photolysis energy. However, this was limited by the
need to ensure pseudo-first-order conditions were maintained and to minimize radical-radical
processes. Ensuring the dominant reaction was between OH and the reagent of study, whilst still
being able to accurately measure the initial OH conditions, provided a limit to the maximum
removal rates achievable ($k_{OH,1st} < 30,000 \text{ s}^{-1}$).

### 277  3 Instrument Characterization

Many reactions of atmospheric and combustion interest are studied in the presence of oxygen
leading to the generation of peroxy radicals ($RO_2$). For certain $RO_2$ there is a potential to generate
OH and $HO_2$ on a fast timescale and therefore three well known reactions were chosen to
characterize the instrument, OH and ethane, OH and ethylene, and OH and methanol.
$OH + C_2H_6 \rightarrow H_2O + C_2H_5$ (R6)
$OH + C_2H_4 \rightarrow HOC_2H_4$ (R7)
$OH + CH_3OH \rightarrow H_2O + CH_2OH, CH_2OH + O_2 \rightarrow HO_2 + HCHO$ (fast) (R8a, R9)
$OH + CH_3OH \rightarrow H_2O + CH_3O, CH_3O + O_2 \rightarrow HO_2 + HCHO$ (slow) (R8b, R10)
OH and ethane (R6) gives an assessment of any false yields generated from $RO_2$ and NO from
prototypical alkyl $RO_2$ species that will be formed from many atmospherically relevant reactions.
Ethylene and OH (R7) forms a hydroxy alkyl peroxy radical, the typical $RO_2$ species known to
create interferences in FAGE $HO_2$ detection systems (Fuchs et al., 2011;Hard et al., 1984;Whalley
et al., 2013). Minimizing and understanding the $HO_2$ yield from this reaction allowed for limits to
the selectivity of the instrument to be known. The reaction of OH with methanol is a well



understood reaction; the two isomeric radical products react with oxygen on differing timescales
to generate $HO_2$. Complete conversion of both isomers should yield 100 % $HO_2$.
As discussed in the experimental section, transport effects after the breakup of the sampling
jet mean that rate coefficients measured in the second cell $k_{X,2nd}$ (X = OH or $HO_2$) differ from each
other (transport effects scale with mass) and from those made in the first detection axis ($k_{OH,1st}$).
Pseudo-first-order rate coefficients from the two detection axes were compared to ascertain
whether measurements in the second detection axes can be used to make quantitative kinetic
measurements.
Finally, the layout of the apparatus makes it hard to accurately measure the temperature at
which the reaction occurs; for reactions occurring on a millisecond timescale, the relevant reaction
distance from the sampling pinhole is approximately 0.05 - 0.5 mm. Compared to a conventional
slow flow laser flash photolysis/laser induced fluorescence apparatus, where the reaction volume
is the overlap of the perpendicular photolysis and probe laser beams, it is hard to accurately
position the thermocouple and additionally, any thermocouple located close to the sampling
pinhole may affect the flow into the low pressure system. In addition to the difficulties in correctly
siting a thermocouple, there are additional errors (flow, conduction and radiative) that derive from
measuring the temperature of a flowing gas with a thermocouple. We have therefore performed
additional experiments to determine the temperature based on the well characterized and
temperature sensitive reaction of OH and methane.

### 312   3.1 Interference by $RO_2$ species

Selectivity in measuring $HO_2$ concentrations plays an important role in the viability of detection
methods for monitoring reactions important for atmospheric chemistry. At high pressures, the
reaction of NO with many atmospherically relevant $RO_2$ species in the presence of oxygen induces
$HO_2$ formation. By performing the titration in the low-pressure cell with the NO + $HO_2$ reaction
under 'starved NO' conditions minimized this effect. This premise was validated by measuring the
OH + ethane and OH + ethylene $HO_2$ yields under high oxygen conditions. In our system the
typical oxygen concentrations in the high pressure reactor were varied between $1 \times 10^{16}$ and $5 \times$
$10^{17}$ molecule $cm^{-3}$ which led to concentrations in the low pressure cell of $3 \times 10^{12}$ to $2 \times 10^{15}$
molecule $cm^{-3}$.





The reaction of OH + ethane (R6) under high oxygen conditions permits the rapid
formation of the ethylperoxy radical, which is an $RO_2$ radical that has a typical slow, NO
propagated, route to $HO_2$ formation (R11 – R12).
$CH_3CH_2O_2 + NO \rightarrow CH_3CH_2O + NO_2$                             (R11)
($k_{11, 298 K} = 8.70 \times 10^{-12}$ cm$^3$ molecule$^{-1}$ s$^{-1}$) (Atkinson et al., 2006)
$CH_3CH_2O + O_2 \rightarrow CH_3CHO + HO_2$                              (R12)
($k_{12, 298 K} = 9.48 \times 10^{-15}$ cm$^3$ molecule$^{-1}$ s$^{-1}$) (Atkinson et al., 2006)
Under a variety of NO flows the apparent $HO_2$ yield for the OH + $C_2H_6$ system was $3 \pm 6$ %, which
indicates that for most reactions carried out in our system, chemical transformation by reaction
with NO, was sensitive to $HO_2$ rather than $RO_2$ species, where the $RO_2$ radical was the product of
$O_2$ addition to a simple alkyl radical.
The reaction of ethylene and OH (R7) in the presence of oxygen forms the
hydroxyethylperoxy radical ($HOCH_2CH_2O_2$), and reaction of the $HOCH_2CH_2O_2$ with NO in the
presence of $O_2$ provides a route for the prompt regeneration of OH. For this reaction, an apparent
$HO_2$ yield of $100 \pm 15$ % was observed; however, by varying the concentration of NO added to
low pressure cell (between $5 \times 10^{13}$ and $5 \times 10^{14}$ molecule cm$^{-3}$), the formation rate of OH was
reduced minimizing the apparent yield observed (<70 %) and slowing the observed rate of OH
regeneration (<1000 s$^{-1}$).
$HOCH_2CH_2O_2 + NO \rightarrow HOCH_2CH_2O + NO_2$                    (R13)
($k_{13, 298 K} = 9.00 \times 10^{-12}$ cm$^3$ molecule$^{-1}$ s$^{-1}$) (Atkinson et al., 2006)
$HOCH_2CH_2O \rightarrow CH_2O + HOCH_2$                             (R14)
($k_{14, 298 K} = 1.3 \times 10^5$ s$^{-1}$) (Orlando et al., 1998)
$CH_2OH + O_2 \rightarrow CH_2O + HO_2$                                (R9)
($k_{9, 298 K} = 9.60 \times 10^{-12}$ cm$^3$ molecule$^{-1}$ s$^{-1}$) (Atkinson et al., 2006)
For test reagents which can generate radicals similar to hydroxyethylperoxy, our instrument will
detect both $HO_2$ and $RO_2$ with some selectivity to $HO_2$.




### 3.2 OH + methanol

To verify the accuracy of the method for determining $HO_2$ yields the reaction of OH and methanol (R8) was examined. The branching ratio for the $\alpha$ abstraction to yield $CH_2OH$ (R8a) reported by the IUPAC evaluation and based on the experimental data of McCaulley et al. (1989), is $\alpha = (85 \pm 8)\%$ at room temperature with the methoxy yield as $(15 \pm 8)\%$. Reaction R8 was studied at room temperature with the reaction being initiated by the photolysis of $H_2O_2$ at 248 nm. In the presence of low concentrations of oxygen ($< 1 \times 10^{16}$ molecule cm$^{-3}$), the $\alpha$ abstraction still leads to prompt formation of $HO_2$ via R9, but R10, $CH_3O + O_2$, occurs on a much longer timescale and is not observed under these conditions. The observed $HO_2$ yield, $(87 \pm 10)\%$ (first row of Table 1) gives the fraction of reaction R8 forming $CH_2OH$ and the value is consistent with the IUPAC evaluation.

**Table 1.** $HO_2$ yields from the reaction of OH with $CH_3OH$ with varying $[O_2]$ carried out 295 K. Errors given as 2 $\sigma$.

| $[O_2]/$ molecule cm$^{-3}$ | HO$_2$ Yield (%) | | | | Average HO$_2$ Yield (%) |
|---|---|---|---|---|---|
| | Expt 1 | Expt 2 | Expt 3 | Expt 4 | |
| $<1 \times 10^{16}$ | 90 | 89 | 79 | 88 | $(87 \pm 10)$ |
| $>2 \times 10^{18}$ | 93 | 94 | 100 | 99 | $(97 \pm 6)$ |

When higher concentrations of oxygen are used, the timescale for $HO_2$ production from reaction R10 decreases and now both abstraction channels lead to $HO_2$ detection in our apparatus. The resulting observed yield (second row of Table 1) is consistent with 100% conversion of OH to $HO_2$. The reproduction of the literature $HO_2$ yields from the reaction of OH with methanol under varying $[O_2]$ demonstrates that the instrument can accurately measure $HO_2$ yields with good precision. It has additionally been demonstrated that the instrument had sufficient accuracy and precision to assign the branching ratios for differing abstraction channels when it was possible to separate the channels by the timescale for $HO_2$ generation.




### 3.3 Assessment of transport effects on observed kinetics

Due to the differing conditions in the two detection regions, the kinetics observed at the first
detection axis, where OH LIF was performed in the jet-expanded gas, and in the second detection
region, where LIF is performed 15 cm downstream from the pinhole after the breakdown of the
jetting gas, will be treated separately. For validating the accuracy of the OH kinetics in the first
cell, the reactions of OH and methane ($CH_4$) (Dunlop and Tully, 1993), OH and ethylene ($C_2H_4$)
(Atkinson et al., 1982; Tully, 1983) were studied. The high accuracy and precision of this system
for measuring OH kinetics has further been demonstrated in a recent publication on the reaction
of OH and isoprene ($C_5H_8$) (Medeiros et al., 2018).
$OH + CH_4 \rightarrow H_2O + CH_3$ (R15)
$OH + C_2H_4 \rightarrow HOC_2H_4$ (R7)
$OH + C_5H_8 \rightarrow HOC_5H_8$ (R16)
When these reactions were carried out at room temperature the expected bimolecular rate
coefficients could be reproducibly accurately measured for observed rate coefficients less than
150,000 s$^{-1}$.
Studies on the reaction of OH and ethylene at room temperature and 2.2 bar, shown in
Figure 5, gave a value of $k_7 = (8.33 \pm 0.16) \times 10^{-12}$ cm$^3$ molecule$^{-1}$ s$^{-1}$ (2σ errors) which matched
well with literature high pressure limits for OH and ethylene; where a direct pulsed laser photolysis
laser induced fluorescence study by Tully (1983) gave $k_7 = (8.47 \pm 0.24) \times 10^{-12}$ cm$^3$ molecule$^{-1}$
s$^{-1}$, and a relative rate study by Atkinson et al. (1982) found $k_7 = (8.11 \pm 0.37) \times 10^{-12}$ cm$^3$
molecule$^{-1}$ s$^{-1}$. However, for pseudo-first order rate coefficients above ~150000 s$^{-1}$, there was no
longer a linear dependence of the rate coefficient with reagent concentration; transport effects are
becoming significant even for OH detection in the jetting region.



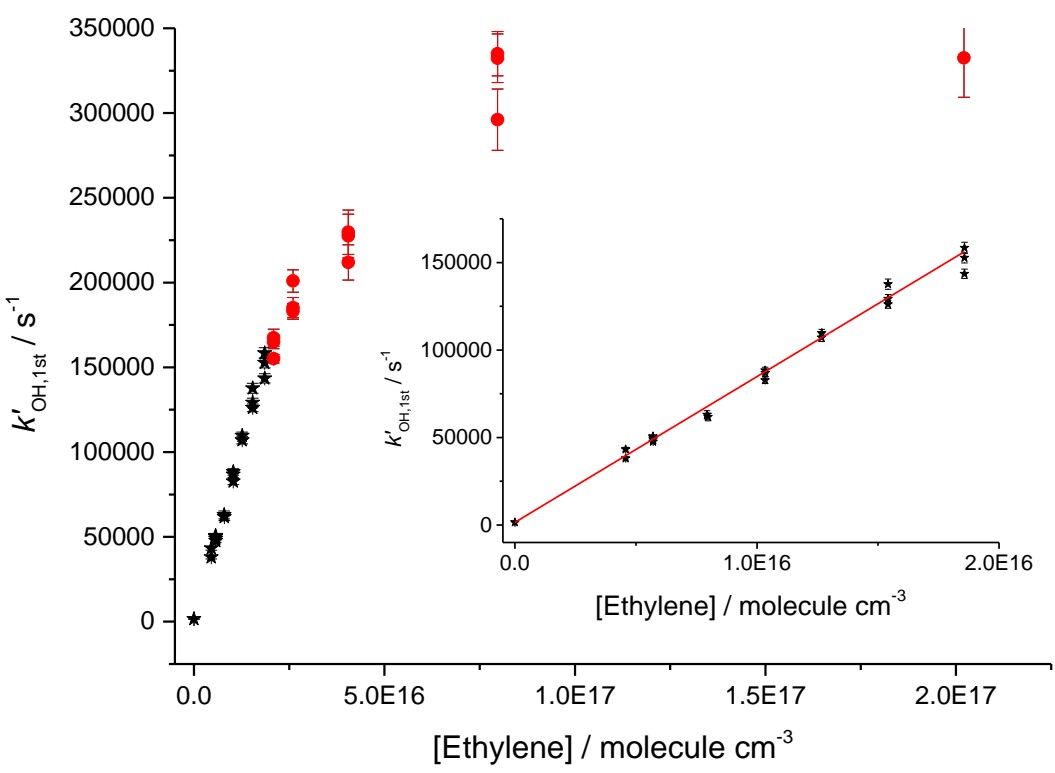


**Figure 5.** Bimolecular plot of the pseudo-first-order rate coefficient at the first detector, $k'_{OH,1st}$, vs ethylene concentration. The figure demonstrates a linear relationship below ~150,000 s$^{-1}$ (see inset for detail in linear region) but with increasing curvature, due to transport effects at higher values of $k'_{OH,1st}$. Black stars symbolize where $k'_{OH,1st}$ was linear with [$C_2H_4$], red circles where $k'_{OH,1st}$ showed greater than 5% deviation from linearity.


The OH traces detected in the second cell deviated from those observed from the first cell,

as shown in Figures 3 and 4. There is understandably an increased time delay from time zero (the
photolysis laser pulse) to arrival of OH radicals at the second detection axis due to the increased
distance travelled after sampling (> 150 mm versus < 5 mm). Additionally, the arrival of OH to
the second axis is spread out further in time due to transport issues relating to non-linear flow
conditions at the breakdown of the jet, and the arrival of the OH being affected by its velocity
distribution (Moore and Carr, 1977;Taatjes, 2007;Baeza-Romero et al., 2012). Figure 6 shows a
plot of observed OH rate coefficient from the first detection axis ($k_{OH,1st}$) against observed rate
coefficient from the second axis ($k_{OH,2nd}$). For values of $k_{OH}$ below 2500 s$^{-1}$ it was possible to


accurately assign an expected OH removal rate for reactions observed in the second cell ($k_{OH,2nd}$)
given the observed OH kinetics at the first detection axis ($k_{OH,1st}$). This is useful to compare the
kinetics of OH removal and HO$_2$ production.

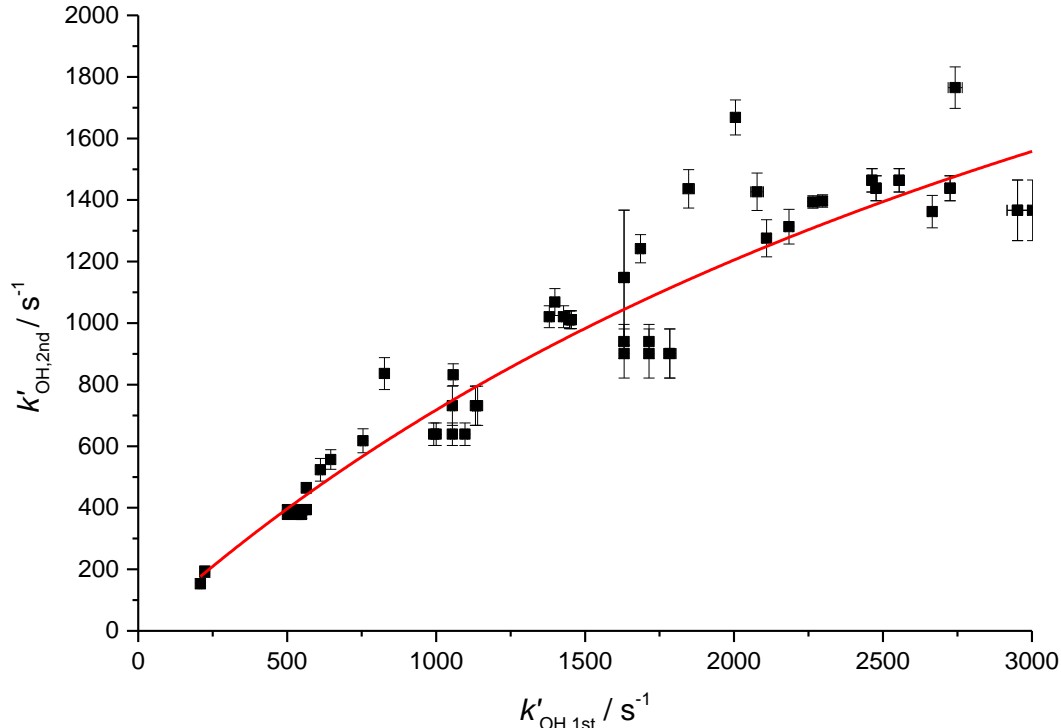


**Figure 6.** Relationship between the observed rate coefficient observed in the first cell ($k_{OH,1st}$) and
the observed OH removal rate in the second cell ($k_{OH,2nd}$). The difference is non-linear but a simple
fit to this could be used to assign removal rates to traces observed in the second cell below 2,500
s$^{-1}$. The red line is the simplified fit of the form, $y = A * (1 - e^{-b*x})$, where A was some limit
value above which no increase in measured rate coefficient would be observed.


As the observed kinetics in the second cell are significantly affected by the velocity
distribution of the species being detected, there is again a deviation between the observed kinetics
expected from the measurement of the OH radicals loss and the kinetics for HO$_2$ formation due to
the differing masses of OH and HO$_2$. Figure 7 shows the pseudo-first order rate coefficients for
OH removal ($k'_{OH,2nd}$) and HO$_2$ production ($k'_{HO2,2nd}$) determined at the second detection axis,
plotted against the OH removal at the first detection axis. The two fits to the data shown in Figure





7 had a ratio of gradients concordant with the root of the masses for $HO_2$ and OH, $0.60 \pm 0.14$
versus the expected relationship of 0.73. As with Figure 6, it is possible to establish a calibration
graph that relates the kinetics of $HO_2$ production at the second detection axis with the primary
kinetics taking place in the high pressure reactor. This means that the timescale over which the
$HO_2$ yield was observed could be assigned and therefore it is possible to attribute $HO_2$ yields to
fast processes, intramolecular $RO_2$ decompositions or $R + O_2$ reactions, or to slower radical-radical
reactions.

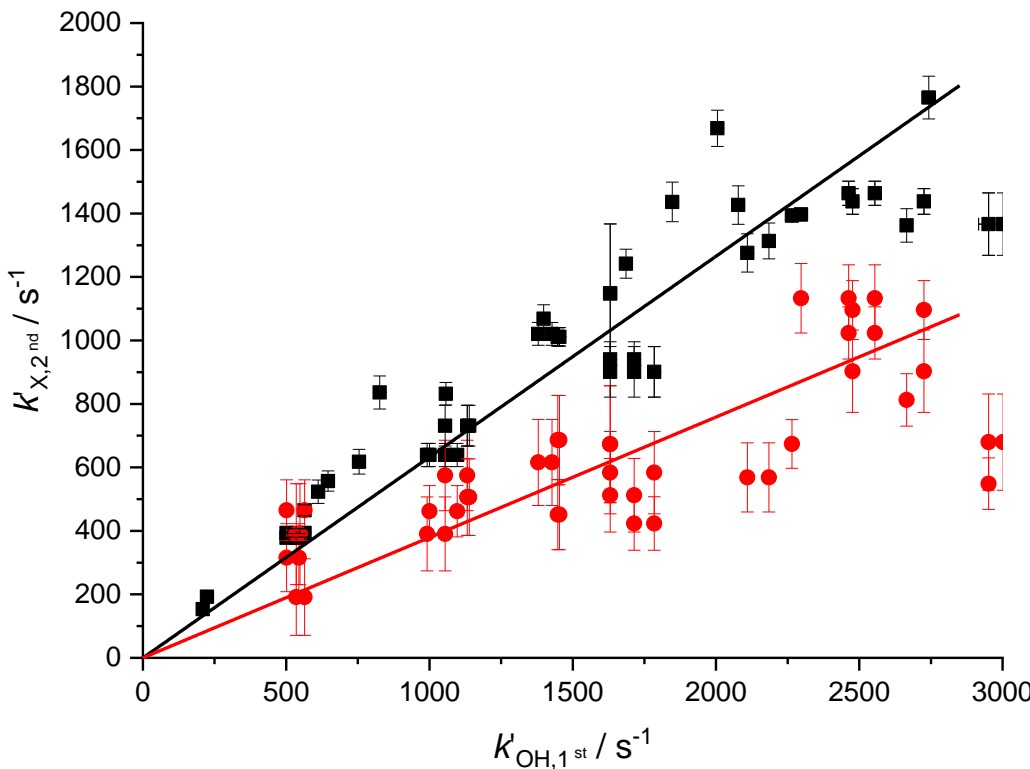


**Figure 7.** Relationship between the pseudo-first-order rate coefficient for OH loss observed in the
first cell ($k'_{OH,1st}$) and the observed rate coefficients in the second cell ($k'_{X,2nd}$ where X = OH or
$HO_2$) a non-linear fit can be used to assign removal rates and $HO_2$ formation rates to traces
observed in the second cell below 2,500 $s^{-1}$.



### 3.4 Temperature corrections

It is difficult to know the exact temperature at the pinhole as introducing a thermocouple close to the region will affect the flows and cannot be used in routine operation. A translatable thermocouple was passed along the axis of the high-pressure reactor over a variety of temperatures and showed that the temperature of the gas at the pinhole varies with axial location. In addition, radial profiles showed that in our system there was insufficient heating length to achieve uniform radial heating of the laminar gas. From the axial measurements it was observed that slower flow rates (< 5 SLM) allowed for reduced axial temperature gradients. However, these measurements showed that the only manner to achieve an even thermal profile would be a static cell.

A permanently seated thermocouple was placed perpendicularly to the flow, close to the sampling region, measurements from this thermocouple were then compared with temperature assignments from the reaction of OH and methane using the temperature dependence assigned by Dunlop and Tully (1993). This was performed over a range of heater settings and flows to allow for temperature assignment. This method was also applied to a standard low-pressure cell where the flows can be reduced to slow enough flows that thermocouple measurements could accurately define the temperature to verify the method. Additionally, the well-determined OH + ethylene adduct formation equilibrium was measured over a range of temperatures to provide an additional verification of the temperature assignment.

The method to assign a temperature from the reaction of OH and methane used the pseudo-first order rate coefficients ($k'_{OH,1st}$) measured at the first detection axis over a range of added methane flows. An estimate of the temperature was made from the thermocouple measurement, this estimated temperature was used, along with the pressure in the reactor, to calculate the added methane concentration. Comparing the predicted pseudo-first-order rate coefficient that this estimated concertation provided using the literature value of $k_{OH+CH4}$ (Dunlop and Tully, 1993) to the measured rate coefficient produced a difference for each point. The estimated temperature was then iteratively changed to minimize the difference between estimated and measured rate coefficients. For this minimum value, the difference between thermocouple measurement and actual temperature was tabulated against the voltage setting for the heater. A parameterization of voltage versus temperature difference was used to estimate the temperature of the reactor for experiments where no OH and methane measurements were performed and has been shown to





reliably predict the temperature of the reactor within 7 K when measurements have been made
subsequently.

To assess the axial temperature gradients in the gas sampled through the pinhole over the

timescales of reactions measured, OH and methane rate coefficients were measured using
photolysis of water at 193 nm as a source of OH. Using water photolysis allowed for low removal
rates of OH by precursor and assignment of OH and methane over a range of pseudo-first-order
rate coefficients ($k'_{OH,1st}$) $100 - 40000$ s$^{-1}$ as shown in Figure 8. This was performed at two
temperatures (505, 680 K), and the slope of observed OH removal rate coefficients against
concentration of methane appeared linear over the full range for both temperatures, thus verifying
that over the distances sampled within experimental timeframes there is a minimal temperature
gradient.

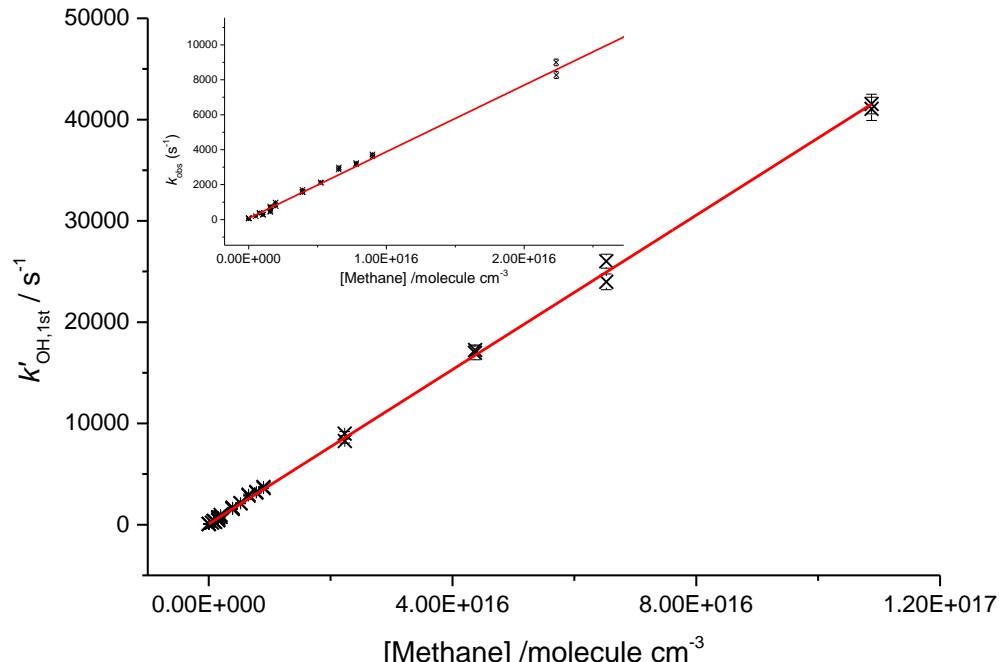


**Figure 8.** Bimolecular plot for the reaction of OH and methane at 680 K, 1760 Torr using 193 nm
photolysis of water as an OH precursor. Here the inset shows that even at removal rates < 1000 s$^{-1}$
the plot is still linear, indicating that within the measured experimental timescales there is little
deviation in temperature.



**4 Determination of Site Specific Rate Coefficients for the Reaction of OH with n-butanol**


The branching ratios for the sites of OH attack on n-butanol, as presented in Scheme 2, are of
significance to the modelling of the ignition delay times for n-butanol (Agbro et al., 2017).
Abstractions at the $\alpha$ and OH positions are chain terminating reactions at low temperatures due to
the formation of the relatively inert $HO_2$ radical, and abstraction at the $\beta$ site leads to chain
propagation, through OH recycling. The new instrument permitted determination of the attack at
the $\alpha$ and $\beta$ sites; attack at the $\alpha$ site leads to prompt $HO_2$ formation in the presence of $O_2$; at
elevated temperatures biexponential fits to non-single exponential OH loss traces in the absence
of $O_2$ (due to decomposition of the $\beta$ hydroxy radical to OH and iso-butene) allowed for attack at
the $\beta$ site to be measured.

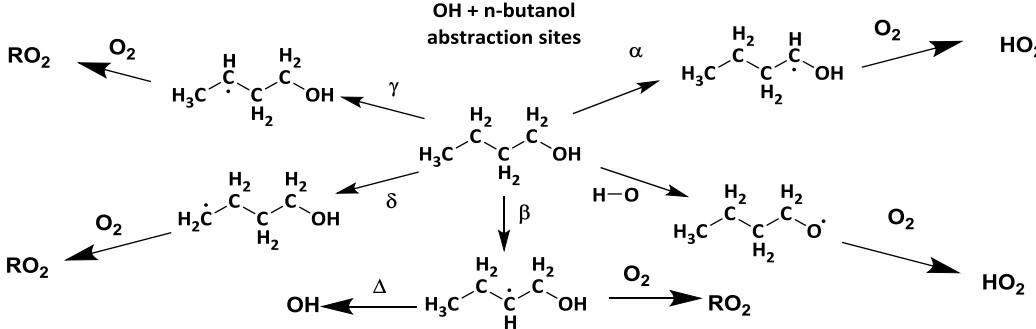


**Scheme 2.** The potential sites for OH abstractions in the oxidation of n-butanol. Of particular
importance to low temperature combustion is the ratio of $\alpha$ to $\beta$ branching fractions where $\alpha$ attack
leads to chain inhibition and beta to chain propagation.

**4.1 Room temperature OH kinetics**
At room temperature under pseudo-first-order conditions ($[OH] < 3 \times 10^{12}$ molecule cm$^{-3}$, [n-
butanol] $> 1.5 \times 10^{14}$ molecule cm$^{-3}$), the OH loss traces recovered from the first detection axis
from the jet expanded gas corresponded closely with single exponential decays. These decays
relate to the overall loss process for the reaction of OH with n-butanol:
OH + n-C$_4$H$_9$OH $\rightarrow$ H$_2$O + isomers of C$_4$H$_9$O (R1)



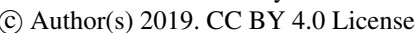


The resulting rate coefficients were plotted against the concentration of butanol, in the presence of
both low and high oxygen, as shown in Figure 9 (low oxygen $< 5 \times 10^{15}$ molecule cm$^{-3}$, high
oxygen  $1.2 \times 10^{19}$ molecule cm$^{-3}$), where $k_{obs} = k_1 \times$ [butanol], giving a resultant bimolecular
removal rate of $k_1 = (9.24 \pm 0.40) \times 10^{-12}$ cm$^3$ molecule$^{-1}$ s$^{-1}$ under low oxygen conditions, and $k_1$
$= (9.23 \pm 0.15) \times 10^{-12}$ cm$^3$ molecule$^{-1}$ s$^{-1}$ under high oxygen conditions.

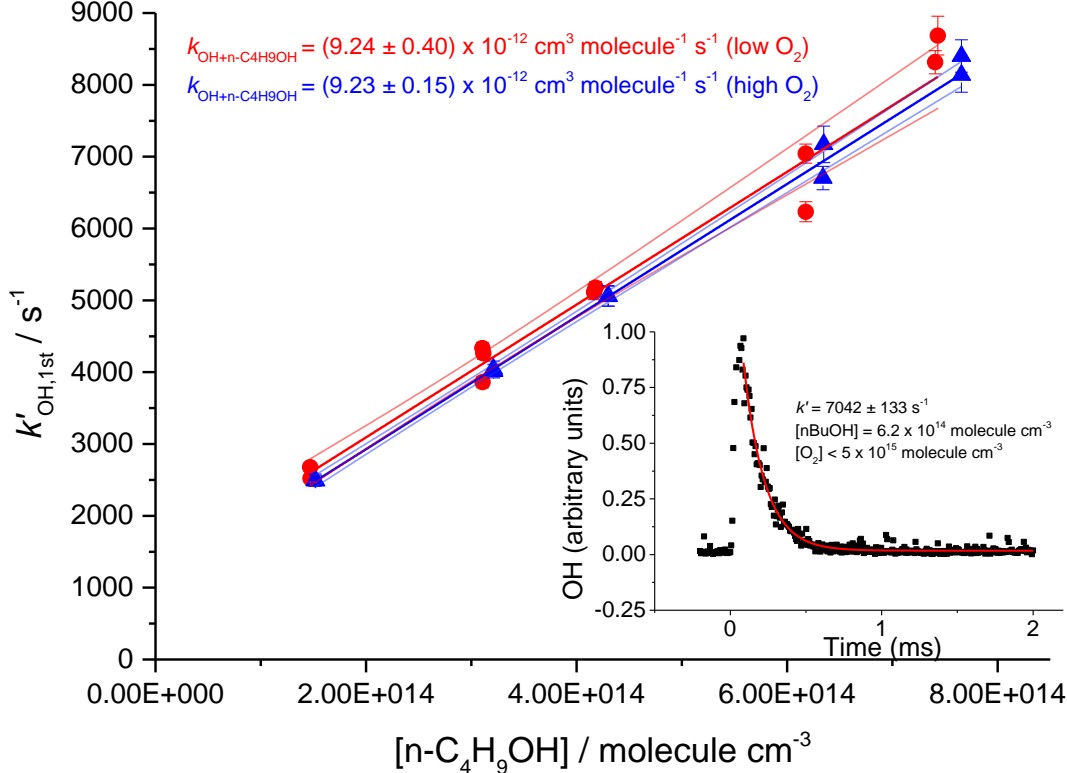


**Figure 9.** Plots $k'_{OH,1st}$ against the concentration of butanol, at two oxygen concentrations, $< 5 \times$
$10^{15}$ molecule cm$^{-3}$ and $1.2 \times 10^{19}$ molecule cm$^{-3}$. Bimolecular rate coefficients were taken from
the slopes as $(9.24 \pm 0.40) \times 10^{-12}$ cm$^3$ molecule$^{-1}$ s$^{-1}$ under low oxygen conditions (red circles with
95 % confidence limits), and $(9.23 \pm 0.15) \times 10^{-12}$ cm$^3$ molecule$^{-1}$ s$^{-1}$ under high oxygen conditions
(blue triangles with 95 % confidence limits). The inset shows a typical OH temporal profile at the
first detection axis.


The good agreement between the measured rate coefficients with varying [O$_2$] verifies that,
as expected under our experimental conditions at room temperature, the R radical formed from the
β abstraction does not undergo fragmentation to OH and but-1-ene. The resultant combined data
gives an overall 293 K bimolecular rate coefficient for OH and n-butanol of $k_1 = (9.24 \pm 0.21) \times$





$10^{-12}$ cm$^3$ molecule$^{-1}$ s$^{-1}$, which is in excellent agreement with the recent work of McGillen et al.
(2013) of $k_{1,296} = (9.68 \pm 0.75) \times 10^{-12}$ cm$^3$ molecule$^{-1}$ s$^{-1}$.

**4.2 Room temperature HO$_2$ results**

Experiments were carried out in high oxygen conditions ($3 \times 10^{17}$ – $1.2 \times 10^{18}$ molecule cm$^{-3}$), at
296 – 303 K, and high pressures (1800 – 2000 Torr) of N$_2$ bath gas using photolysis of hydrogen
peroxide at two different wavelengths (248 nm and 266 nm), and the resulting HO$_2$ yields are
shown in Table 2. The resulting HO$_2$ yield was determined to be ($58 \pm 7$) % at 266 nm, and ($55 \pm$
12) % at 248 nm. As there is no significant variation in the yield with laser wavelength or power,
we can treat the data in Table 2 as 12 independent estimates of the yield, giving an averaged HO$_2$
yield of 57% with a standard error (95%) of 6%. Therefore under the experimental conditions
(pressure >1800 Torr, [O$_2$] > $3 \times 10^{17}$ molecule cm$^{-3}$), the HO$_2$ yield, which originates from OH
attack at the α abstraction site, was ($57 \pm 6$) %, with a minor contribution from abstraction from
the hydroxyl group. The yield assigned is in good agreement with McGillen et al. (2013) 57%, and
Cavalli et al. (2002) $52 \pm 7$ %.

**Table 2.** HO$_2$ yields from experiments carried out at room temperature (293 – 298 K) with reaction
initiated by photolysis of H$_2$O$_2$ at 248 nm and 266 nm.

| Laser wavelength/nm | HO$_2$ Yield (%) | | | | | | | Average HO$_2$ Yield (%) |
|---|---|---|---|---|---|---|---|---|
| | Expt 1 | Expt 2 | Expt 3 | Expt 4 | Expt 5 | Expt 6 | Expt 7 | |
| 266 | $61 \pm 7$ | $54 \pm 4$ | $46 \pm 5$ | $56 \pm 7$ | $54 \pm 7$ | $67 \pm 10$ | $66 \pm 6$ | $58 \pm 7$ |
| 248 | $63 \pm 2$ | $68 \pm 2$ | $48 \pm 5$ | $52 \pm 5$ | $49 \pm 5$ | | | $55 \pm 12$ |


Experiments were carried out with photolysis at 266 nm and at a variety of laser energies at 248
nm, the yields remained consistent with photolysis wavelength and power. Varying the laser power
did alter the profiles of the HO$_2$ traces recovered; the growth rates remained unaffected but the
tails changed; decreasing laser power slowed the removal rate of HO$_2$ (from greater than 100 s$^{-1}$
to under 10 s$^{-1}$) showing that radical-radical processes are the main source of HO$_2$ loss from the
system. If radical-radical reactions were an important source of any observed HO$_2$ yield changing
laser power would have altered the HO$_2$ yield and additionally the HO$_2$ growth kinetics.





### 4.3 Higher temperature – HO$_2$ yield and OH recycling

The R radical formed from abstraction at the β site (CH$_3$CH$_2$CHCH$_2$OH) can regenerate OH and form butene, Scheme 2, in the absence of added oxygen. This process was not observed at ambient temperatures (293 – 305 K) but at elevated temperatures, 616 K – 657 K, the OH loss observed at the first detection axis was no longer well described by a single exponential loss process, Figure 10. The non-exponential decays formed were due to OH being returned following decomposition of the β R radical. Biexponential fits to the recycling traces gave the fraction of OH returned (Medeiros et al. 2018), with an average β branching fraction of (23 ± 4)%, Table 3.

**Table 3.** OH recycling and HO$_2$ yields from experiments carried out under elevated temperatures (> 600 K) all experiments were carried out with photolysis at 248 nm.

| Temperature (K) | Fraction of OH returned (%) | Observed HO$_2$ Yield (%) |
|---|---|---|
| 616 | 24.2 ± 4.1 | 54 ± 4 |
| 622 | 24.4 ± 4.9 | |
| 636 | 25.7 ± 5.6 | |
| 657 | 18.1 ± 4.0 | |

The HO$_2$ yield measured at an elevated temperature (616 K), where OH recycling was also observed, was 54 ± 4 %  (Table 3) which is within error of the value (57 ± 6 %) measured at room temperature (293 K), although it is not possible to partition the HO$_2$ yield between α and OH abstraction. Over the temperature range tested the branching ratio for OH attack at the α position is therefore also likely to remain unchanged. With the sum of the α and β sites contributing (78 ± 4)%, at 616 K, the remainder of the abstraction (~22%) occurs at the δ and γ sites. These results are in excellent agreement of the product study of Cavalli et al. (2002) which found (52 ± 7)% α from the butanal product yield and (23 ± 4) % β from the propanal yield using FTIR detection and the site specific analysis by McGillen et al. (57 % α and 26% β). The product study of Hurley et al. (2009) found 44 ± 4 % α and 19 ± 2 % β values which are lower than our experimental values but are within the combined error ranges. However, it should be noted that the β branching fraction of 23 ± 4 % measured in this study was obtained at elevated temperatures, 616 – 657 K.




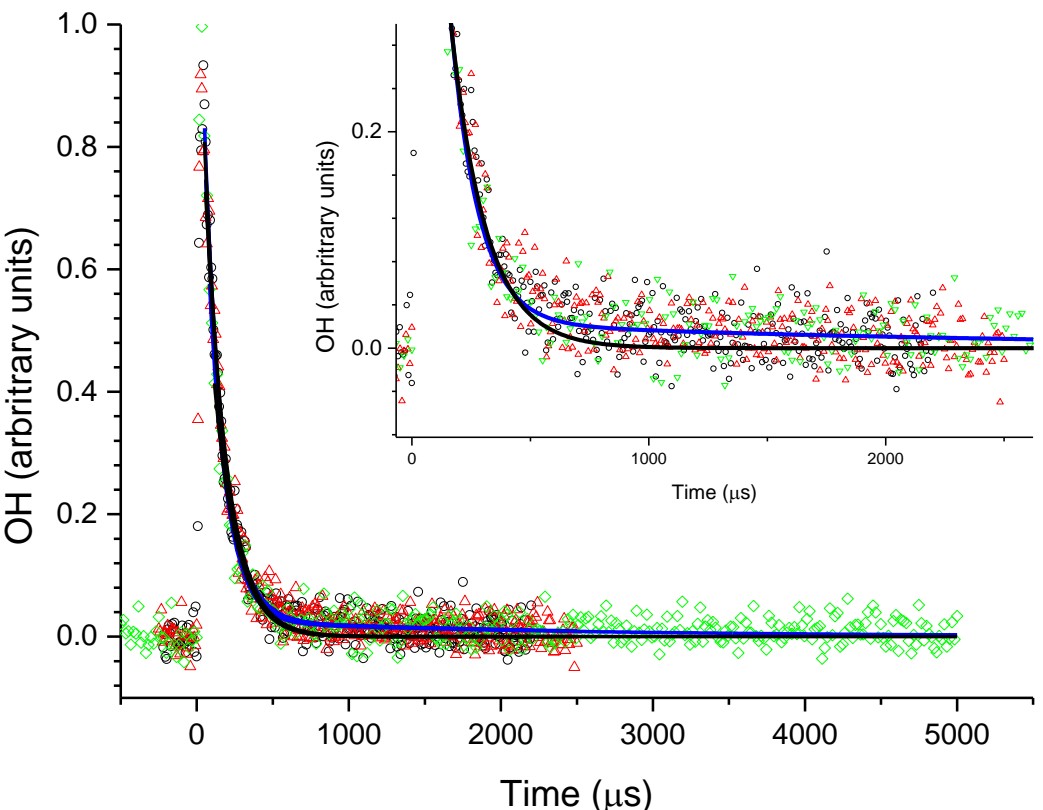

581

**Figure 10.** An example of the OH signal collected at the first detection axis for the reaction of OH
with n-butanol ($[nBuOH] \approx 1.4 \times 10^{15}$ molecule cm$^{-3}$, at 616 K  The black line represents the least
squares fits to an exponential decay ($k'_{1,1st} = (6780 \pm 380)$ s$^{-1}$), with the blue line representing a bi-
exponential fit ($k_{1,biexp,1st} = (8190 \pm 180)$ s$^{-1}$)).


**5 Summary**

The use of $H_2O_2$ as an OH precursor has been shown to provide a reliable method of internally
characterizing our system for $HO_2$ yield detection. Interferences that could arise from using this
precursor for $HO_2$ detection have been accounted for, and the presence of water that the $H_2O_2$
precursor introduces has no effect on the sensitivity of the LIF method, unlike IR absorption
methods.



It has been demonstrated that this instrument can reliably assign $HO_2$ yields and
simultaneously measure OH kinetics, even under conditions of high temperatures and high oxygen
concentrations, which could be challenging for other detection systems. Such conditions are
important for exploring key combustion chemistry reactions, and for converting slow
atmospherically relevant processes to the microsecond timescales required to minimize secondary
or heterogeneous chemistry.
By performing reactions under low NOx and low radical densities ( $<1 \times 10^{13}$ molecule
$cm^{-3}$), $HO_2$ yields formed on fast timescales ($> 300$ $s^{-1}$) can be assigned to direct $HO_2$ channels or
reactions of alkyl (R) radicals with oxygen. Whilst some time-resolution is lost at the $HO_2$ detector,
sufficient time-resolution is retained in order to separate varying sources of $HO_2$, for example the
two channels leading to $HO_2$ production in the $OH/CH_3OH/O_2$ system (Section 3.2) or from
unwanted secondary chemistry.
For particular reactions, illustrated in this paper by the study of OH with methanol and
butanol in the presence of oxygen, the simultaneous measurement of OH kinetics and $HO_2$ yields
can provide important site-specific information. In other systems, the onset of $HO_2$ formation could
allow for the assignment of new channels becoming important within complex mechanism,
potentially allowing for verifying the onset of OOQOOH chemistry within OH regeneration
processes.

### 6 Acknowledgements

A studentship from NERC for T.H. Speak is gratefully acknowledged.

### 7 Author Contributions

THS undertook most of the experimental measurements and contributed to the first draft of the
manuscript. DS provided input into experimental design and analysis of transport effects. MAB
and PWS lead the project and completed the manuscript.





**8 Competing Interests**

The authors declare that they have no conflicts of interest.

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
