# Peer review of "Manuscript under review for journal Atmos. Meas. Tech."

_Atmospheric Measurement Techniques, 2019_

## Referee Comment (RC1) · Anonymous Referee #2 · 14 May 2019

The authors describe an experimental apparatus to determine HO2 yields and OH reaction kinetics in a pump-probe flow-tube experiment. The paper is suitable for publication in AMT after addressing the following points: P1 L19/20: As written now, the statement only verifies the OH kinetics. P2 L41/42: It would be useful to show the explicit reactions. P6 L148: What was the repetition rate of the laser? P8 L182: Which range and which resolution was used for the delay between photolysis and detection?

P8 L201: Could the authors show here or elsewhere that the chemistry stopped, when the air entered the low pressure cells or what the influence on the measurement was, if not?

P11 L246-252: Could the authors give some numbers for the correction?

[Figure]

P11 L253-268: The description could be extended by giving more details what exactly is calculated and how calibration numbers are derived. Is an absolute OH calibration of the cells needed for this approach? If so, how was this achieved?

P14 L346: What are the consequences for not so well-known systems? Is there a strategy how to estimate the RO2 fraction in the signal or at least to know, if RO2 influenced the yield?

P15 L357: Could the authors give numbers of the timescales? What fraction of HO2 from R10 would be still seen?

P15 L370: I kindly disagree with this statement. The yield is the difference between the HO2 yields from both experiments has a large error. The value is (10+/-11)% applying error propagation. What would be the additional uncertainty due to potential RO2 interferences and the fraction of HO2 from R10 (see comment above)?

P16 Section 3.3: The description would benefit from a discussion about the reproducibility of these effects and their impact on the accuracy of results for experiments.

Figure 6/7: The authors should make clear, which experiments are shown in these figures.

Table 3: The table is not correctly displayed.

The authors might somewhere discuss the approach used in Nehr et al., PCCP, 2002 to determine HO2 yields.

General remark to the figures: It would be easier to work with legends instead of descriptions in the captions.
* * *

---

## Referee Comment (RC2) · Anonymous Referee #1 · 24 May 2019

The paper describes a new experimental set-up allowing the measurement of time-resolved $HO_2$ traces. This is a very interesting approach, and the paper should be published. However, I have, besides some minor remarks, a major concern: you do not take into account any secondary radical-radical reaction with the argument, that your radical concentrations are low enough. I do not agree with this point, even though it is not always easy to get enough details from the manuscript to judge. So my comment is based on your statement page 6, that the typical initial OH concentration is between 2e11 and 5e13 $cm^{-3}$. In the below graph are shown two simulations with $[OH]_0$ = 1e12 and $H_2O_2$ = 5e14 (left) and $[OH]_0$ = 1e13 and $[H_2O_2]$ = 1e15 (right graph). The blue symbols show the simple model OH + $H_2O_2$ → $HO_2$ + $H_2O$, while the green symbols include on top the reaction of OH + $HO_2$ → $H_2O$ + $O_2$ with 1e-10 $cm^3s^{-1}$.

[Figure]

It can very clearly be seen that even under the relatively low initial radical concentration of 1e12 (which is at your lower end) already the $HO_2$ yield is not 100% anymore, situation gets much worse with 1e13 OH: only 60% of the initial OH is converted to $HO_2$. This has also an influence on the OH decay rate, as well as on the retrieved $HO_2$ rise time (both get faster). This "problem" has been discussed in detail by Assaf et al, JPCA 2016, when using this system to retrieve the OH absorption cross section. In your case not taking into account secondary chemistry will lead to an overestimation of the $HO_2$ yield. Of course taking into account this chemistry is possibly only if you know the absolute initial OH concentration. Maybe you did some experiments were you varied the photolysis energy? Because this would give you an idea if secondary reactions are important or not under your conditions.

In the case of the OH + $CH_3OH$ experiments, secondary chemistry might play a role as well. Very recently, Assaf et al (PCCP, 20, 10660, 2018) have measured the rate constant of $CH_3O$ + $HO_2$ and $CH_3O$ + $CH_3O$, both have found to be very fast (1.1e-10 and 7e-11 $cm^3s^{-1}$). The result is that even under moderate high initial radical concentrations, some $CH_3O$ will react away before it is converted into $HO_2$. You find a yield in good agreement with literature, either your initial radical concentration

are at the lower end of the indicated range, or maybe the internal calibration, tenting to overestimate the yield, makes up for this underestimation. Please give more information on the estimated initial radical concentration for the different experiments and check, if your systems are really free from secondary chemistry. In any case, before I can agree to the sentence that your instrument can accurately measure $HO_2$ yields, I would like to see a more detailed discussion on possible secondary chemistry.

Figure 3 : the black squares are difficult to distinguish from the blue triangle. Better chose other symbols or other colors.

Figure 7: Who is who? I guess red is $HO_2$ and black is OH? What was the reaction system in Figure 7 and what was the estimated initial radical concentration? Because from the above model, one would expect a faster $HO_2$ decay compared to OH decay if secondary reactions are taken into account (2003 $s^{-1}$ for OH against 2596 $s^{-1}$ for $HO_2$ in the example of the right graph above).

Figure 10: what are the different colored symbols? Different experiments? Or is the blue line a fit to different data points?

---

## Short Comment (SC1) · 12 Jun 2019

Atmos. Meas. Tech. Discuss., doi:10.5194/amt-2019-164-RC1, 2019 © Author(s) 2019. "A New Instrument for Time Resolved Measurement of HO2 Radicals" by Thomas H. Speak et al. Anonymous Referee #2

*The authors describe an experimental apparatus to determine $HO_2$ yields and OH reaction kinetics in a pump-probe flow-tube experiment. The paper is suitable for publication in AMT after addressing the following points:*

**Comment 1:**

*P1 L19/20: As written now, the statement only verifies the OH kinetics.*

"As an application of the new instrument, the reaction of OH with n-butanol has been studied at 293 and 616 K. The bimolecular rate coefficient at 293 K, $(9.24 \pm 0.21) \times 10^{-12}$ cm$^3$ molecule$^{-1}$ s$^{-1}$ (18 , 19) is in good agreement with recent literature, verifying that this instrument can both measure $HO_2$ yields and accurate OH kinetics."

**Response:**

Agreed, the wording is unclear and is now worded as (removing mention of yields):

'The bimolecular rate coefficient at 293 K, $(9.24 \pm 0.21) \times 10^{-12}$ cm$^3$ molecule$^{-1}$ s$^{-1}$ is in good agreement with recent literature, verifying that this instrument can measure accurate OH kinetics.'

Validation of the $HO_2$ yields is emphasised later in the abstract, where we now state:

'Direct observation of the $HO_2$ product in the presence of oxygen has allowed the assignment of the α-branching fractions $(0.57 \pm 0.06)$ at 293 K and $(0.54 \pm 0.04)$ at 616 K), again in good agreement with the literature;'

**Comment 2:**

*P2 L41/42: It would be useful to show the explicit reactions.*

"whereas abstraction at other sites leads to alkylperoxy radical (C4H9O2)formation with varying  fractions of the RO2 forming alkoxy radicals, and subsequently HO2 (McGillen et al., 2013) on a  longer timescale."

**Response:**

The decision has been made to use Scheme 2 to provide clarity on these reactions (listing all reactions takes up too much space) and this is now referenced here.

**Comment 3:**

*P6 L148: What was the repetition rate of the laser?*

"The photolysis of the OH precursor, H2O2, at 248 nm (Lambda Physik, Compex 200  operated using KrF) or 266 nm (frequency quadrupled Nd-YAG output, Quantel, Q-smart 850) initiated the chemistry."

**Response:**

Experiments were carried out with repetition rates varied between 0.5 and 10 Hz, varying the repetition rates within this range did not affect the observed OH kinetics and $HO_2$ yields. As a result of this repetition rate independence, in general experiments were carried out at 5 and 10 Hz for 248 nm and 10 Hz for 266 nm. However, for each reaction the assumption of repetition rate independence was verified by performing an experiment at 1 Hz in addition to the higher repetition rate experiments.

An explicit description of this is now included. In addition, this will be discussed clearly in the description of the work done to check for the effect of any radical radical processes that will be included at the behest of Reviewer 1.

Revised wording:

'The photolysis of the OH precursor, $H_2O_2$, at 248 nm (Lambda Physik, Compex 200 operated using KrF at 1 or 5 Hz) or 266 nm (frequency quadrupled Nd-YAG output, Quantel, Q-smart 850 at 1 or 10 Hz) initiated the chemistry. No significant difference was noted in the kinetics or yields as a function of laser repetition rate.'

**Comment 4:**

*P8 L182: Which range and which resolution was used for the delay between photolysis and detection?*

**Response:**

Typical experimental traces contained $200 - 300$ data points sampling the experimental time frame which were in the range $50 - 190,000$ microseconds at 5 Hz, and $50 - 95,000$ microseconds at 10 Hz. Typical delays between the pump and probe lasers were on the microsecond timescale with control of these timings in the high nanoseconds.

This is now described more clearly in the final manuscript. 'A delay generator (BNC DG535) was used to vary the delay (time resolution ~10 ns) between the photolysis and probe laser, facilitating generation of time profiles of the OH concentration. The traces, typically $200 - 300$ data points and ranging in time from ~50 μs – 20 ms, were scanned through multiple times (5 – 20) and the signal at each time point was averaged, giving high precision OH loss traces.'

**Comment 5:**

*P8 L201: Could the authors show here or elsewhere that the chemistry stopped, when the air entered the low-pressure cells or what the influence on the measurement was, if not?*

**Response:**

The pressure drop ($1600 - 0.5$ Torr) from the high pressure to the low pressure cell will reduce the rate of bimolecular reactions proportionally. It is acknowledged that the density in the jet itself is higher ($10 - 60$ Torr). However, rate constants can be measured within 1-2 % of the literature (Medieros et al. *J. Phys. Chem. A* 2018), and minimal quenching of the OH LIF signal over a wide range of added oxygen show that chemistry occurring within the jet is minimal. For unimolecular reactions, the temperature change from the expansion ensures that the rates of these processes are slowed significantly.

**Comment 6:**

*P11 L246-252: Could the authors give some numbers for the correction?*

" For reactions carried out where a reagent was added in addition to the H2O2, the resulting ratios can be compared with those from the calibration reaction to allow assignment of an observed HO2 yield. To assign the HO2 yield from the test reaction required accounting for secondary HO2 production in the high-pressure reactor, from OH + H2O2 and photolysis processes. From the known rate coefficients, it was possible to calculate the fraction of OH reacting with the H2O2 and hence the expected contribution to the HO2 signal. Photolytic production of HO2 was accounted for by measuring the observed HO2 signal in the absence of any H2O2."

**Response:**

OH and RH was typically kept 10 to 20 times faster than OH and hydrogen peroxide and from this using the kinetics of the respective reactions the fraction of OH that reacted with the precursor could simply be accounted for. In general, this accounted for 5 to 10 % of the observed $HO_2$ signal and this is accounted for explicitly within our analysis.

Where photolysis of the reagents leads to HCO or H in the presence of oxygen this provides an additional source of $HO_2$. The observed signal in the absence of the OH precursor was subtracted from signal in the presence of the OH precursor. For the reactions included in this paper there was no observed photolysis of the reagents.

$RO_2 + RO_2$ can be a source of $HO_2$ however under our experimental conditions this forms too slowly (2-80 $s^{-1}$) to provide a significant increased $HO_2$ yield.

'From the known rate coefficients, it was possible to calculate the fraction of OH reacting with the $H_2O_2$ (typically 5 – 10%) and hence the expected contribution to the $HO_2$ signal.'

**Comment 7:**

*P11 L253-268: The description could be extended by giving more details what exactly is calculated and how calibration numbers are derived. Is an absolute OH calibration of the cells needed for this approach? If so, how was this achieved?*

**Response:**

No, absolute concentrations are not required. A reference reaction with a known $HO_2$ yield is used and then compared to the reaction under study, as stated in Lines 240 to 252 of the original manuscript.

This has been validated by comparing two reference reactions OH and $H_2O_2$ and OH and $CH_3OH$ with high oxygen, 100 % $HO_2$. By proving that $H_2O_2$ and $CH_3OH$ give the same $HO_2$ yields we can simply use OH and $H_2O_2$ on its own.

**Comment 8:**

*P14 L346: What are the consequences for not so well-known systems? Is there a strategy how to estimate the RO2 fraction in the signal or at least to know, if RO2 influenced the yield?*

"For test reagents which can generate radicals similar to hydroxyethylperoxy, our instrument will detect both HO2 and RO2 with some selectivity to HO2."

**Response:**

As with all FAGE $HO_2$ detectors this instrument will not fully discriminate between $RO_2$ and $HO_2$. As highlighted in this section the instrument cannot be described as exclusively an HO2 detector.

Discriminating $RO_2$ from $HO_2$ relies on the requirement for multiple NO reactions for OH formation in the case of $RO_2$ radicals. By varying the [NO] and knowing the $RO_2$ --> OH kinetics can identify where $RO_2$ is being detected, under these conditions defining $HO_2$ yields becomes complex as is described in (Nehr et al. PCCP 2011).

'For test reagents which can generate radicals similar to hydroxyethylperoxy, our instrument will detect both $HO_2$ and $RO_2$ with some selectivity to $HO_2$. Potential $RO_2$ interference can be tested by examining the '$HO_2$' yield as a function of added [NO].'

**Comment 9:**

*P15 L357: Could the authors give numbers of the timescales? What fraction of HO2 from R10 would be still seen?*

"the α abstraction still leads to prompt formation of HO2 via R9, but R10, CH3O + O2, occurs on a much longer timescale"

**Response:**

R9 had a formation rate of over 50,000 $s^{-1}$ compared with R10, which had a formation rate of approximately 10 $s^{-1}$. Even for the slowest OH and methanol reactions carried out the yield was assigned before 5 millisecond, under these conditions less than 10 percent of this channel, will have formed $HO_2$ under the measured timescale. With the $HO_2$ peak being retrieved from the biexponential fit, it is likely that this contribution would have been lower than at under 3 percent of this channel being titrated to $HO_2$ at the point at which the peak $HO_2$ signal was observed.

**Comment 10:**

*P15 L370: I kindly disagree with this statement. The yield is the difference between the HO2 yields from both experiments has a large error. The value is (10+/-11)% applying error propagation. What would be the additional uncertainty due to potential RO2 interferences and the fraction of HO2 from R10 (see comment above)?*

"It has additionally been demonstrated that the instrument had sufficient accuracy and precision to assign the branching ratios for differing abstraction channels when it was possible to separate the channels by the timescale for HO2 generation."

**Response:**

This statement was based purely on the result of simple statistical significance at the 95 % confidence level. When a Welch's t test (*Biometrika*, 1947) was performed on the low and high oxygen measurements (($0.87 \pm 0.10$), ($0.97 \pm 0.06$), with errors given as twice the standard deviation) a t value of 3.41 was derived with 5.10 degrees of freedom, for a 2 tailed test this gave

a p value of 0.0184 which is statistically significant at the 98 % confidence level, this result was not significant at the 99 % level.

Further experiments were carried out on this reaction with respect to the question posed by Reviewer 1, and through this work the upper yield has now been revised to $98 \pm 2$ % where the error is given as 2 sigma/sqrt(n).

When the new revised value for the high oxygen measurements is used the p value returned is 0.0156, again significant at the 98 % level but not the 99 % level.

In the low oxygen experiments the $[O_2] < 1E15$ cm$^{-3}$ and therefore $CH_3O + O_2 < 20$ s$^{-1}$. When the $HO_2$ was assigned < 5% of $CH_3O$ would have been titrated to $HO_2$.

'The resulting observed yield (second row of Table 1) is consistent with 100% conversion of OH to $HO_2$ and is statistically different from the low oxygen measurements based on a Welch t-test at the 98% level.'

**Comment 11:**

P16 Section 3.3: The description would benefit from a discussion about the reproducibility of these effects and their impact on the accuracy of results for experiments.

"3.3 Assessment of transport effects on observed kinetics"

**Response:**

Kinetics measured in the jet require no corrections.

Any kinetics measured in a FAGE expansion outside of the jet itself are subject to transport. The kinetics measured on the second detector are always slower (10% under 250 s$^{-1}$ 50 % at 2500 s$^{-1}$), this deviation can be corrected for via Figures 6 and 7. The effect of transport on kinetics has been discussed in detail by Stone et al. (R.S.I 2016) and by Taatjes (Int. J. Chem. Kinet. 2007) for transport in the jetting gas. The effects on transport when sampling from high to low pressures is also described in detail in Baeza-Romero et al. (Int. J. Chem. Kinet. 2011), references to these works will be included in the final manuscript.

**Comment 12:**

Figure 6/7: The authors should make clear, which experiments are shown in these figures.

**Response:**

These experiments were measurements of OH and $H_2O_2$ over varied $H_2O_2$ concentrations. Legends have now been included, and the description has been updated to provide the experimental detail.

**Comment 13:**

Table 3: The table is not correctly displayed.

**Response:**

Thank you, the table will display fine in a final print, the issue is line numbers have displayed over the table as oppose to at the side of the page as would normally be expected. The table itself seems correct though.

**Comment 14:**

The authors might somewhere discuss the approach used in Nehr et al., PCCP, 2002 to determine HO2 yields.

**Response:**

Nehr et al PCCP 2011, Phys. Chem. Chem. Phys., 2011, 13, 10699–10708 on HO2 from OH and benzene using a modified OH reactivity instrument does include an interesting and thorough description of assigning $HO_2$ yields from OH initiated reactions. A discussion of the technique of Nehr et al. and a comparison to the method used in this work will be included.

**Comment 15:**

General remark to the figures: It would be easier to work with legends instead of descriptions in the captions.

**Response:**

Use of appropriate legends supported by detailed figure captions will indeed improve the overall readability of this work. See above comments.

---

## Short Comment (SC2) · 12 Jun 2019

**First comment 1:**

"However, I have, besides some minor remarks, a major concern: you do not take into account any secondary radical-radical reaction with the argument, that your radical concentrations are low enough. I do not agree with this point, even though it is not always easy to get enough details from the manuscript to judge. So my comment is based on your statement page 6, that the typical initial OH concentration is between 2e11 and 5e13 cm-3. In the below graph are shown two simulations with [OH]0 = 1e12 and H2O2 = 5e14 (left) and [OH]0 = 1e13 and [H2O2] = 1e15 (right graph). The blue symbols show the simple model OH + H2O2 ◊ HO2 + H2O, while the green symbols include on top the reaction of OH + HO2 ◊ H2O + O2 with 1e-10 cm3 s -1 . t / s [OH, HO2] 0.000 0.002 0.004 0.006 0.008 0.010 0 5.0×101 1 1.0×101 2 HO2 OH with secondary reactions OH w/o secondary reactions Y0 Plateau K oh 1.001e+012 = 0.0 887.4 x 10.000e+011 = 0.0 848.7 HO2_ini k_slow HO2_sec k_fast ho2 = 0.0 3.015 8.923e+011 949.9 xy = 0.0 -0.03680 9.998e+011 849.2 HO2 t / s [OH, HO2] 0.000 0.001 0.002 0.003 0.004 0.005 0 5.0×101 2 1.0×101 3 Y0 Plateau K oh 1.002e+013 = 0.0 2003 x 9.998e+012 = 0.0 1687 HO2_ini k_slow HO2_sec k_fast ho2 = 0.0 20.16 6.379e+012 2596 xy = 0.0 -0.09674 9.993e+012 1691 ho2 oh x It can very clearly be seen that even under the relatively low initial radical concentration of 1e12 (which is at your lower end) already the HO2 yield is not 100% anymore, situation gets much worse with 1e13 OH: only 60% of the initial OH is converted to HO2. This has also an influence on the OH decay rate, as well as on the retrieved HO2 rise time (both get faster). This "problem" has been discussed in detail by Assaf et al, JPCA 2016, when using this system to retrieve the OH absorption cross section. In your case not taking into account secondary chemistry will lead to an overestimation of the HO2 yield.

Of course taking into account this chemistry is possibly only if you know the absolute initial OH concentration. Maybe you did some experiments were you varied the photolysis energy? Because this would give you an idea if secondary reactions are important or not under your conditions. In the case of the OH + CH3OH experiments, secondary chemistry might play a role as well. Very recently, Assaf et al (PCCP, 20, 10660, 2018) have measured the rate constant of CH3O + HO2 and CH3O + CH3O, both have found to be very fast (1.1e-10 and 7e-11 cm3 s -1 ). The result is that even under moderate high initial radical concentrations, some CH3O will react away before it is converted into HO2. You find a yield in good agreement with literature, either your initial radical concentration are at the lower end of the indicated range, or maybe the internal calibration, tending to overestimate the yield, makes up for this underestimation. Please give more information on the estimated initial radical concentration for the different experiments and check, if your systems are really free from secondary chemistry. In any case, before I can agree to the sentence that your instrument can accurately measure HO2 yields, I would like to see a more detailed discussion on possible secondary chemistry."

**Response**

Thank you for this very pertinent question. Prior to submission many checks for radical-radical effects were made by varying the repetition rate and photolysis laser power, and no observed differences were seen in the $HO_2$ yields. We had based our statement on the empirical observations rather than a review of the possible secondary chemistry, however, your careful review and analysis, does suggest that we ought to see a significant difference. The literature does imply we should see a change in $HO_2$ yield between $OH + H_2O_2$ and $OH + CH_3OH/O_2$ (n.b. note that the same high $[O_2]$ is used in both experiments). As the $[OH]$ is increased we should get an ~50 % yield when $[OH] \sim= 3E13$ cm$^{-3}$ and $OH + H_2O_2$ is used as the $OH \rightarrow HO_2$ conversion reaction. Also the observed OH removal kinetics of the $OH + H_2O_2$ reaction should increase with $[OH]$; >20% faster when $[OH] \sim=3E13$ cm$^{-3}$. Because of this inconsistency of our result with the literature, we have carried a number of new experiments, where the $[OH]$ is

varied over a greater range, varying pump laser power from 0.5-60 mJ cm$^{-2}$. The take home message is that we cannot reproduce the literature, and our HO$_2$ yields / kinetics for the reaction OH + H$_2$O$_2$ are close to unchanged over all [OH] from 2E11 up to 5E13 molecule cm$^{-3}$.

We are in agreement with the reviewer as to the implications of the literature. From a model (as detailed by the reviewer), when [OH]0 = 1E13 cm$^3$, the HO$_2$ yield from OH + H$_2$O$_2$ is about 50% compared to when a large excess of methanol is added, see Fig 1.

[Figure]

*Figure 1 A simulation of the expected HO$_2$ yields for reaction of 9E12 OH with 1E15 H$_2$O$_2$ with 6E18 O$_2$, in the presence and absence of 1E16 methanol. Where the removal of OH by reaction with the HO$_2$, OH were included, and accounting for the loss of HO$_2$ via reaction with HO$_2$, OH, CH$_3$O and diffusion.*

The crucial reaction in attenuating the HO$_2$ yield in the OH + H$_2$O$_2$ reaction is OH + HO$_2$. According to the literature the HO$_2$ yield will only be close to unity when [OH] < 1E12 cm$^{-3}$. Also, it is noted that when OH + HO$_2$ is significantly occurring the OH + H$_2$O$_2$ kinetics will be significantly faster than the literature. Based on these predictions we have done further experiments.

By comparing HO$_2$ yield when CH$_3$OH(O$_2$) is present we can assign yields without knowing the absolute radical concentration; it is wholly reasonable to assign the HO$_2$ yield in the presence of sufficient CH$_3$OH(O$_2$) as100%. Comparing the HO$_2$ yield from OH and methanol in the presence of high oxygen, to the yield in the absence of methanol allowed for the assignment of HO$_2$ yields from the reaction of OH with hydrogen peroxide.

The literature predicts a large decrease in the $HO_2$ yield from hydrogen peroxide as the $[OH]_0$ is increased, see Fig 1. Below is our yield for $HO_2$ from $OH + H_2O_2$ and is compared to when a large amount of $CH_3OH(O_2)$ is added. It is clear that no attenuation of the $HO_2$ yield is observed in our system. Many other experiments were carried out as $[OH]_0$ was varied over a factor of ~300, and the $HO_2$ yield from all the experiments was the same, within error, for $OH + H_2O_2$ compared to when $CH_3OH(O_2)$ is added. The $[H_2O]$ and $[CH_3OH]$ in the system is too small for significant complexation to $HO_2$. Our experiments assign yields as 100% (101 ± 7 at 1E11 [OH], 101 ± 2 at 3E12 [OH] and 102 ± 3 at 3E13 [OH]).

[Figure]

*Figure 2 $HO_2$ growth profiles collected with 2.5E13 cm$^{-3}$ [OH], 6E18 cm$^{-3}$ $O_2$, 2.7E15 $H_2O_2$ in the presence and absence of 2.5E16 $CH_3OH$.*

Also, the impact of the $HO_2 + OH$ on the observed $OH + H_2O_2$ rate constant is to make it significantly faster as the initial [OH] is increased. From our literature model, measurable changes in the rate constant should be observed as [OH] is increased >20 % for 1 Hz experiments; in experiments carried out at 10 Hz where there is $HO_2$ present from the previous laser flash at time zero, this should lead to the observation of an increase in the OH removal rate by up to 50 %. The precision of the system means that we can readily see changes in the rate constant to ~ 1% when initial OH radical concentrations are >1E12 cm$^{-3}$. The results are summarised is the graph below.

[Figure]

*Figure 3 Expected and observed OH removal rates with 2.7E15 cm$^{-3}$ H$_2$O$_2$ and 1-60 mJ cm$^{-2}$ photolysis energy at 248 nm and 10Hz.*

Again, this kinetics test versus [OH] demonstrates that under our conditions HO$_2$ + OH is having little impact on the OH + H$_2$O$_2$ reaction. The measurable increase in the figure 3 (6.8%) can be assigned to OH + OH ($\approx$1E-11 cm$^3$ s$^{-1}$, at 1600 Torr).

We recognise that our results are in contradiction with the literature rate coefficient for HO$_2$ + OH. The IUPAC literature value is 1.1E-10 cm$^3$ s$^{-1}$. In order to reconcile our experiments we require this rate coefficient to be < 1E-11 cm$^3$ s$^{-1}$. However, our result is wholly consistent with the previous paper on the reaction of OH with H$_2$O$_2$ (Wine et al. 1981  J. Chem. Phys). In Wine et al. the removal kinetics were not perturbed by additional HO$_2$ added to the system. In this work, with additional [HO$_2$] ~ 1E13 added, no measurable change in the OH + H$_2$O$_2$ was observed. This result is in agreement with our present study. We note that Wine et al study used flash photolysis study, as used in our present study. Most literature assignments on HO + HO$_2$ were carried out in low pressure, flow tubes; very different conditions. The flash photolysis is less prone to interference.

"Figure 3 : the black squares are difficult to distinguish from the blue triangle. Better chose other symbols or other colours."

**Response:**

In part, that these are hard to distinguish is due to these traces showing no evidence of back diffusion of NO into the region where the OH is probed in the first detection axis. However, we will adjust this figure to highlight the different traces better.

**Comment 3:**

"Figure 7: Who is who? I guess red is HO2 and black is OH? What was the reaction system in Figure 7 and what was the estimated initial radical concentration? Because from the above model, one would expect a faster HO2 decay compared to OH decay if secondary reactions are taken into account (2003 s -1 for OH against 2596 s-1 for HO2 in the example of the right graph above)."

A legend will provide for clarity in this figure, and the OH concentrations were 1-3 E12 cm$^{-3}$. The experimental detail has now been included in the description.

Please see the response above for comment on observations of secondary reactions.

Comment 4:

Figure 10: what are the different colored symbols? Different experiments? Or is the blue line a fit to different data points?

**Response:**

The red fit is an exponential fit to the data, the blue fit is a multi-exponential fit that allows for assignment of the returned OH. The three colours of symbols are merely three different time scans of the same experimental conditions to allow for correct assignment of both the fast and slow loss processes.  A legend will add clarity to this figure.

---

## Author Comment (AC1) · 31 Jul 2019

**First comment 1:**

"However, I have, besides some minor remarks, a major concern: you do not take into account any secondary radical-radical reaction with the argument, that your radical concentrations are low enough. I do not agree with this point, even though it is not always easy to get enough details from the manuscript to judge. So my comment is based on your statement page 6, that the typical initial OH concentration is between 2e11 and 5e13 cm-3. In the below graph are shown two simulations with [OH]0 = 1e12 and H2O2 = 5e14 (left) and [OH]0 = 1e13 and [H2O2] = 1e15 (right graph). The blue symbols show the simple model OH + H2O2 ◊ HO2 + H2O, while the green symbols include on top the reaction of OH + HO2 ◊ H2O + O2 with 1e-10 cm3 s -1 . t / s [OH, HO2] 0.000 0.002 0.004 0.006 0.008 0.010 0 5.0×101 1 1.0×101 2 HO2 OH with secondary reactions OH w/o secondary reactions Y0 Plateau K oh 1.001e+012 = 0.0 887.4 x 10.000e+011 = 0.0 848.7 HO2_ini k_slow HO2_sec k_fast ho2 = 0.0 3.015 8.923e+011 949.9 xy = 0.0 -0.03680 9.998e+011 849.2 HO2 t / s [OH, HO2] 0.000 0.001 0.002 0.003 0.004 0.005 0 5.0×101 2 1.0×101 3 Y0 Plateau K oh 1.002e+013 = 0.0 2003 x 9.998e+012 = 0.0 1687 HO2_ini k_slow HO2_sec k_fast ho2 = 0.0 20.16 6.379e+012 2596 xy = 0.0 -0.09674 9.993e+012 1691 ho2 oh x It can very clearly be seen that even under the relatively low initial radical concentration of 1e12 (which is at your lower end) already the HO2 yield is not 100% anymore, situation gets much worse with 1e13 OH: only 60% of the initial OH is converted to HO2. This has also an influence on the OH decay rate, as well as on the retrieved HO2 rise time (both get faster). This "problem" has been discussed in detail by Assaf et al, JPCA 2016, when using this system to retrieve the OH absorption cross section. In your case not taking into account secondary chemistry will lead to an overestimation of the HO2 yield.

Of course taking into account this chemistry is possibly only if you know the absolute initial OH concentration. Maybe you did some experiments were you varied the photolysis energy? Because this would give you an idea if secondary reactions are important or not under your conditions. In the case of the OH + CH3OH experiments, secondary chemistry might play a role as well. Very recently, Assaf et al (PCCP, 20, 10660, 2018) have measured the rate constant of CH3O + HO2 and CH3O + CH3O, both have found to be very fast (1.1e-10 and 7e-11 cm3 s -1 ). The result is that even under moderate high initial radical concentrations, some CH3O will react away before it is converted into HO2. You find a yield in good agreement with literature, either your initial radical concentration are at the lower end of the indicated range, or maybe the internal calibration, tending to overestimate the yield, makes up for this underestimation. Please give more information on the estimated initial radical concentration for the different experiments and check, if your systems are really free from secondary chemistry. In any case, before I can agree to the sentence that your instrument can accurately measure HO2 yields, I would like to see a more detailed discussion on possible secondary chemistry."

**Response**

Thank you for this very pertinent question. Prior to submission many checks for radical-radical effects were made by varying the repetition rate and photolysis laser power, and no observed differences were seen in the $HO_2$ yields. We had based our statement on the empirical observations rather than a review of the possible secondary chemistry, however, your careful review and analysis, does suggest that we ought to see a significant difference. The literature does imply we should see a change in $HO_2$ yield between $OH + H_2O_2$ and $OH + CH_3OH/O_2$ (n.b. note that the same high $[O_2]$ is used in both experiments. As the $[OH]$ is increased we should get an ~50 % yield when $[OH] \sim= 3E13$ cm$^{-3}$ and $OH + H_2O_2$ is used as the $OH \rightarrow HO_2$ conversion reaction. Also the observed OH removal kinetics of the $OH + H_2O_2$ reaction should increase with $[OH]$; >20% faster when $[OH] \sim=3E13$ cm$^{-3}$. Because of this inconsistency of our result with the literature, we have carried a number of new experiments, where the $[OH]$ is

varied over a greater range, varying pump laser power from 0.5-60 mJ cm$^{-2}$. The take home message is that we cannot reproduce the literature, and our HO$_2$ yields / kinetics for the reaction OH + H$_2$O$_2$ are close to unchanged over all [OH] from 2E11 up to 5E13 molecule cm$^{-3}$.

We are in agreement with the reviewer as to the implications of the literature. From a model (as detailed by the reviewer), when [OH]0 = 1E13 cm$^3$, the HO$_2$ yield from OH + H$_2$O$_2$ is about 50% compared to when a large excess of methanol is added, see Fig 1.

[Figure]

*Figure 1 A simulation of the expected HO$_2$ yields for reaction of 9E12 OH with 1E15 H$_2$O$_2$ with 6E18 O$_2$, in the presence and absence of 1E16 methanol. Where the removal of OH by reaction with the HO$_2$, OH were included, and accounting for the loss of HO$_2$ via reaction with HO$_2$, OH, CH$_3$O and diffusion.*

The crucial reaction in attenuating the HO$_2$ yield in the OH + H$_2$O$_2$ reaction is OH + HO$_2$. According to the literature the HO$_2$ yield will only be close to unity when [OH] < 1E12 cm$^{-3}$. Also, it is noted that when OH + HO$_2$ is significantly occurring the OH + H$_2$O$_2$ kinetics will be significantly faster than the literature. Based on these predictions we have done further experiments.

Also note that by comparing HO$_2$ yield when CH$_3$OH(O$_2$) is present we can assign yields without knowing the absolute radical concentration; it is wholly reasonable to assign the HO$_2$ yield in the presence of sufficient CH$_3$OH(O$_2$) as100%.

The literature predicts a large decrease in the HO$_2$ yield from hydrogen peroxide as the [OH]$_0$ is increased, see Fig 1. Below is our yield for HO$_2$ from OH + H$_2$O$_2$ and  is compared to when a large amount of CH$_3$OH(O$_2$) is added. It is clear that no attenuation of the HO$_2$ yield is observed

in our system. Many other experiments were carried out as $[OH]_0$ was varied over a factor of ~300, and the $HO_2$ yield from all the experiments was the same, within error, for $OH + H_2O_2$ compared to when $CH_3OH(O_2)$ is added. The $[H_2O]$ and $[CH_3OH]$ in the system is too small for significant complexation to $HO_2$. Our experiments assign yields as 100% (99 ± 7 at 1E11 [OH], 99 ± 2 at 3E12 [OH] and 98 ± 3 at 3E13 [OH]).

[Figure]

*Figure 2 $HO_2$ growth profiles collected with 2.5E13 $cm^{-3}$ [OH], 6E18 $cm^{-3}$ $O_2$, 2.7E15 $H_2O_2$ in the presence and absence of 2.5E16 $CH_3OH$.*

Also, the impact of the $HO_2 + OH$ on the observed $OH + H_2O_2$ rate constant is to make it significantly faster as the initial [OH] is increased. From our literature model, measurable changes in the rate constant should be observed as [OH] is increased >20 % for 1 Hz experiments; in experiments carried out at 10 Hz where there is $HO_2$ present from the previous laser flash at time zero, this should lead to the observation of an increase in the OH removal rate by up to 50 %. The precision of the system means that we can readily see changes in the rate constant to ~ 1%. The results are summarised is the graph below.

[Figure]

*Figure 3 Expected and observed OH removal rates with 2.7E15 cm$^{-3}$ H$_2$O$_2$ and 1-60 mJ cm$^{-2}$ photolysis energy at 248 nm and 10Hz.*

Again, this kinetics test versus [OH] demonstrates that under our conditions HO$_2$ + OH is having little impact on the OH + H$_2$O$_2$ reaction. The measurable increase in the figure 3 (6.8%) can be assigned to OH + OH (=1E-11 cm$^3$ s$^{-1}$, at 1600 Torr).

We recognise that our results are in contradiction with the literature rate coefficient for HO$_2$ + OH. The IUPAC literature value is 1.1E-10 cm$^3$ s$^{-1}$. In order to reconcile our experiments we require this rate coefficient to be < 1E-11 cm$^3$ s$^{-1}$. However, our result is wholly consistent with the previous paper on the reaction of OH with H$_2$O$_2$ (Wine et al. 1981  J. Chem. Phys). In Wine et al. the removal kinetics were not perturbed by additional HO$_2$ added to the system. In this work, with additional [HO$_2$] ~ 1E13 added, no measurable change in the OH + H$_2$O$_2$ was observed. This result is in agreement with our present study. We note that Wine et al study used flash photolysis study, as used in our present study. Most literature assignments on HO + HO$_2$ were carried out in low pressure, flow tubes; very different conditions. The flash photolysis is less prone to interference.

We have added new material in the provisional revised manuscript, **lines 382 – 396**.

‘A possible interference that could distort the yield of HO$_2$ is the role of the radical-radical reaction OH + HO$_2$ (Assaf and Fittschen, 2016):

OH + HO$_2$ → H$_2$O + O$_2$        $k_{15}$ = 1.1 × 10$^{-10}$ cm$^3$ molecule$^{-1}$ s$^{-1}$     Atkinson et al. (2006) (R15)

At the low radical concentrations used in many experiments in this work, this reaction could contribute $5 - 10\%$ of the OH loss in an OH + $H_2O_2$ calibration experiment. However, we have looked at the dependence of the $HO_2$ yield from both $OH/H_2O_2$ and from $OH/CH_3OH$, but see no significant effects of secondary radical-radical reaction ($<5\%$) as the calculated $[OH]_0$ is changed from $5 \times 10^{11}$ to $5 \times 10^{12}$ molecule cm$^{-3}$. For the $OH/CH_3OH$ the much larger concentrations of substrate used lead to faster pseudo-first order decays, so radical-radical contribution is significantly reduced. The work of Assaf and Fittschen suggests that a more significant deviation in the OH loss rates, and one that we ought to be to detect given the precision of our data, should be observed. It is possible that our calculations of $[OH]_0$ are over-estimated, but we note that a study of the OH + $H_2O_2$ reaction by Wine et al. (1981), where they specifically looked for the interference on OH decays from R15, could find no evidence for an increase in the loss of OH, when $[HO_2]$ was artificially increased.'

**Comment 2:**

"Figure 3 : the black squares are difficult to distinguish from the blue triangle. Better chose other symbols or other colours."

**Response:**

In part, that these are hard to distinguish is due to these traces showing no evidence of back diffusion of NO into the region where the OH is probed in the first detection axis. We have tried to improve figures by using open symbols in cases.

**Comment 3:**

"Figure 7: Who is who? I guess red is HO2 and black is OH? What was the reaction system in Figure 7 and what was the estimated initial radical concentration? Because from the above model, one would expect a faster HO2 decay compared to OH decay if secondary reactions are taken into account (2003 s -1 for OH against 2596 s-1 for HO2 in the example of the right graph above)."

**A legend has been provided** in the revised manuscript and the OH concentrations were 1-3 E12 cm$^{-3}$. The experimental detail has now been included in the description.

Please see the response above for comment on observations of secondary reactions.

Comment 4:

Figure 10: what are the different colored symbols? Different experiments? Or is the blue line a fit to different data points?

**Response:**

The red fit is an exponential fit to the data, the blue fit is a multi-exponential fit that allows for assignment of the returned OH. The three colours of symbols are merely three different time scans of the same experimental conditions to allow for correct assignment of both the fast and slow loss processes. **Additional material in the caption has added clarity** to this figure.

---

## Author Comment (AC2) · 31 Jul 2019

Atmos. Meas. Tech. Discuss., doi:10.5194/amt-2019-164-RC1, 2019 © Author(s) 2019. "A New Instrument for Time Resolved Measurement of HO2 Radicals" by Thomas H. Speak et al. Anonymous Referee #2

*The authors describe an experimental apparatus to determine $HO_2$ yields and OH reaction kinetics in a pump-probe flow-tube experiment. The paper is suitable for publication in AMT after addressing the following points:*

**Comment 1:**

*P1 L19/20: As written now, the statement only verifies the OH kinetics.*

"As an application of the new instrument, the reaction of OH with n-butanol has been studied at 293 and 616 K. The bimolecular rate coefficient at 293 K, $(9.24 \pm 0.21) \times 10^{-12}$ cm$^3$ molecule$^{-1}$ s$^{-1}$ (18 , 19) is in good agreement with recent literature, verifying that this instrument can both measure HO$_2$ yields and accurate OH kinetics."

**Response:**

Agreed, the wording is unclear and is now worded as (removing mention of yields):

'The bimolecular rate coefficient at 293 K, $(9.24 \pm 0.21) \times 10^{-12}$ cm$^3$ molecule$^{-1}$ s$^{-1}$ is in good agreement with recent literature, verifying that this instrument can measure accurate OH kinetics.'

Validation of the HO$_2$ yields is emphasised later in the abstract, where we now state:

'Direct observation of the HO$_2$ product in the presence of oxygen has allowed the assignment of the α-branching fractions $(0.57 \pm 0.06)$ at 293 K and $(0.54 \pm 0.04)$ at 616 K), again in good agreement with the literature;'

**Comment 2:**

*P2 L41/42: It would be useful to show the explicit reactions.*

"whereas abstraction at other sites leads to alkylperoxy radical (C4H9O2)formation with varying fractions of the RO2 forming alkoxy radicals, and subsequently HO2 (McGillen et al., 2013) on a longer timescale."

**Response:**

The decision has been made to use Scheme 2 to provide clarity on these reactions (listing all reactions takes up too much space) and this has been moved to the relevant part of the manuscript.

**Comment 3:**

*P6 L148: What was the repetition rate of the laser?*

"The photolysis of the OH precursor, H2O2, at 248 nm (Lambda Physik, Compex 200 operated using KrF) or 266 nm (frequency quadrupled Nd-YAG output, Quantel, Q-smart 850) initiated the chemistry."

**Response:**

Experiments were carried out with repetition rates varied between 0.5 and 10 Hz, varying the repetition rates within this range did not affect the observed OH kinetics and $HO_2$ yields. As a result of this repetition rate independence, in general experiments were carried out at 5 Hz for 248 nm and 10 Hz for 266 nm. However, for each reaction the assumption of repetition rate independence was verified by performing an experiment at 1 Hz in addition to the higher repetition rate experiments.

An explicit description of this is now included. In addition, this will be discussed clearly in the description of the work done to check for the effect of any radical radical processes that will be included at the behest of Reviewer 1.

Revised wording (**line 155**):

'The photolysis of the OH precursor, $H_2O_2$, at 248 nm (Lambda Physik, Compex 200 operated using KrF at 1 or 5 Hz) or 266 nm (frequency quadrupled Nd-YAG output, Quantel, Q-smart 850 at 1 or 10 Hz) initiated the chemistry. No significant difference was noted in the kinetics or yields as a function of laser repetition rate.'

**Comment 4:**

*P8 L182: Which range and which resolution was used for the delay between photolysis and detection?*

**Response:**

Typical experimental traces contained 200 – 300 data points sampling the experimental time frame which were in the range 50 – 190,000 microseconds at 5 Hz, and 50 – 95,000 microseconds at 10 Hz. Typical delays between the pump and probe lasers were on the microsecond timescale with control of these timings in the high nanoseconds.

This is now described more clearly in the provisional revised manuscript (**line 189**). 'A delay generator (BNC DG535) was used to vary the delay (time resolution ~10 ns) between the photolysis and probe laser, facilitating generation of time profiles of the OH concentration. The traces, typically 200 – 300 data points and ranging in time from ~50 µs – 20 ms, were scanned through multiple times (5 – 20) and the signal at each time point was averaged, giving high precision OH loss traces.'

**Comment 5:**

*P8 L201: Could the authors show here or elsewhere that the chemistry stopped, when the air entered the low-pressure cells or what the influence on the measurement was, if not?*

**Response:**

The pressure drop (1600 – 0.5 Torr) from the high pressure to the low pressure cell will reduce the rate of bimolecular reactions proportionally. It is acknowledged that the density in the jet itself is higher (10 – 60 Torr). However, rate constants can be measured within 1-2 % of the literature (Medieros et al. *J. Phys. Chem. A* 2018), and minimal quenching of the OH LIF signal over a wide range of added oxygen show that chemistry occurring within the jet is minimal. For unimolecular reactions, the temperature change from the expansion ensures that the rates of these processes are slowed significantly.

*P11 L246-252: Could the authors give some numbers for the correction?*

" For reactions carried out where a reagent was added in addition to the H2O2, the resulting ratios can be compared with those from the calibration reaction to allow assignment of an observed HO2 yield. To assign the HO2 yield from the test reaction required accounting for secondary HO2 production in the high-pressure reactor, from OH + H2O2 and photolysis processes. From the known rate coefficients, it was possible to calculate the fraction of OH reacting with the H2O2 and hence the expected contribution to the HO2 signal. Photolytic production of HO2 was accounted for by measuring the observed HO2 signal in the absence of any H2O2."

**Response:**

OH and RH was typically kept 10 to 20 times faster than OH and hydrogen peroxide and from this using the kinetics of the respective reactions the fraction of OH that reacted with the precursor could simply be accounted for. In general, this accounted for 5 to 10 % of the observed $HO_2$ signal and this is accounted for explicitly within our analysis.

Where photolysis of the reagents leads to HCO or H in the presence of oxygen this provides an additional source of $HO_2$. The observed signal in the absence of the OH precursor was subtracted from signal in the presence of the OH precursor. For the reactions included in this paper there was no observed photolysis of the reagents.

$RO_2$ + $RO_2$ can be a source of $HO_2$ however under our experimental conditions this forms too slowly (2-80 $s^{-1}$) to provide a significant increased $HO_2$ yield.

Revised text (**line 257**)

'From the known rate coefficients, it was possible to calculate the fraction of OH reacting with the $H_2O_2$ (typically 5 – 10%) and hence the expected contribution to the $HO_2$ signal.'

*P11 L253-268: The description could be extended by giving more details what exactly is calculated and how calibration numbers are derived. Is an absolute OH calibration of the cells needed for this approach? If so, how was this achieved?*

**Response:**

No, absolute concentrations are not required. A reference reaction with a known $HO_2$ yield is used and then compared to the reaction under study, as stated in Lines 240 to 252 of the original manuscript.

This has been validated by comparing two reference reactions OH and $H_2O_2$ and OH and $CH_3OH$ with high oxygen, 100 % $HO_2$. By proving that $H_2O_2$ and $CH_3OH$ give the same $HO_2$ yields we can simply use OH and $H_2O_2$ on its own.

*P14 L346: What are the consequences for not so well-known systems? Is there a strategy how to estimate the RO2 fraction in the signal or at least to know, if RO2 influenced the yield?*

"For test reagents which can generate radicals similar to hydroxyethylperoxy, our instrument will detect both HO2 and RO2 with some selectivity to HO2."

**Response:**

As with all FAGE $HO_2$ detectors this instrument will not fully discriminate between $RO_2$ and $HO_2$. As highlighted in this section the instrument cannot be described as exclusively an HO2 detector.

Discriminating $RO_2$ from $HO_2$ relies on the requirement for multiple NO reactions for OH formation in the case of $RO_2$ radicals. By varying the [NO] and knowing the $RO_2$ --> OH kinetics can identify where $RO_2$ is being detected, under these conditions defining $HO_2$ yields becomes complex as is described in (Nehr et al. PCCP 2011).

(**Line 355**)

'For test reagents which can generate radicals similar to hydroxyethylperoxy, our instrument will detect both $HO_2$ and $RO_2$ with some selectivity to $HO_2$. Potential $RO_2$ interference can be tested by examining the '$HO_2$' yield as a function of added [NO].'

**Comment 9:**

*P15 L357: Could the authors give numbers of the timescales? What fraction of HO2 from R10 would be still seen?*

"the α abstraction still leads to prompt formation of HO2 via R9, but R10, CH3O + O2, occurs on a much longer timescale"

**Response:**

R9 had a formation rate of over 50,000 $s^{-1}$ compared with R10, which had a formation rate of approximately 10 $s^{-1}$. Even for the slowest OH and methanol reactions carried out the yield was assigned before 5 millisecond, under these conditions less than 10 percent of this channel, will have formed $HO_2$ under the measured timescale. With the $HO_2$ peak being retrieved from the biexponential fit, it is likely that this contribution would have been lower than at under 3 percent of this channel being titrated to $HO_2$ at the point at which the peak $HO_2$ signal was observed.

The relevant rate coefficients are presented in **line 366**.

**Comment 10:**

*P15 L370: I kindly disagree with this statement. The yield is the difference between the HO2 yields from both experiments has a large error. The value is (10+/-11)% applying error propagation. What would be the additional uncertainty due to potential RO2 interferences and the fraction of HO2 from R10 (see comment above)?*

"It has additionally been demonstrated that the instrument had sufficient accuracy and precision to assign the branching ratios for differing abstraction channels when it was possible to separate the channels by the timescale for HO2 generation."

**Response:**

This statement was based purely on the result of simple statistical significance at the 95 % confidence level. When a Welch's t test (*Biometrika*, 1947) was performed on the low and high oxygen measurements (($0.87 \pm 0.10$), ($0.97 \pm 0.06$), 2 sigma errors) a t value of 3.41 was derived with 5.10 degrees of freedom, for a 2 tailed test this gave a p value of 0.0184 which is statistically significant at the 98 % confidence level, this result was not significant at the 99 % level.

Further experiments were carried out on this reaction with respect to the question posed by Reviewer 1, and through this work the upper yield has now been revised to $99 \pm 4$ % where the error is again given as 2 sigma.

When the new revised value for the high oxygen measurements is used the p value returned is 0.0156, again significant at the 98 % level but not the 99 % level.

In the low oxygen experiments the $[O_2] < 1E15$ cm$^{-3}$ and therefore $CH_3O + O_2 < 20$ s$^{-1}$. When the $HO_2$ was assigned < 5% of $CH_3O$ would have been titrated to $HO_2$.

(**Line 375**)

'The resulting observed yield (second row of Table 1) is consistent with 100% conversion of OH to $HO_2$ ==and is statistically different from the low oxygen measurements based on a Welch t-test at the 95% level==.'

**Comment 11:**

P16 Section 3.3: The description would benefit from a discussion about the reproducibility of these effects and their impact on the accuracy of results for experiments.

"3.3 Assessment of transport effects on observed kinetics"

**Response:**

Kinetics measured in the jet require no corrections.

Any kinetics measured in a FAGE expansion outside of the jet itself are subject to transport. The kinetics measured on the second detector are always slower (10% under 250 s$^{-1}$ 50 % at 2500 s$^{-1}$), this deviation can be corrected for via Figures 6 and 7. The effect of transport on kinetics has been discussed in detail by Stone et al. (R.S.I 2016) and by Taatjes (Int. J. Chem. Kinet. 2007) for transport in the jetting gas. The effects on transport when sampling from high to low pressures is also described in detail in Baeza-Romero et al. (Int. J. Chem. Kinet. 2011), references to these works are included in the provisional revised manuscript (line ***).

**Comment 12:**

Figure 6/7: The authors should make clear, which experiments are shown in these figures.

**Response:**

These experiments were measurements of OH and $H_2O_2$ over varied $H_2O_2$ concentrations. Legends have now been included, and the description has been updated to provide the experimental detail.

**Comment 13:**

Table 3: The table is not correctly displayed.

**Response:**

Thank you, the table will display fine in a final print, the issue is line numbers have displayed over the table as oppose to at the side of the page as would normally be expected. The table itself seems correct though.

**Comment 14:**

The authors might somewhere discuss the approach used in Nehr et al., PCCP, 2002 to determine HO2 yields.

**Response:**

Nehr et al PCCP 2011, Phys. Chem. Chem. Phys., 2011, 13, 10699–10708 on HO2 from OH and benzene using a modified OH reactivity instrument does include an interesting and thorough description of assigning $HO_2$ yields from OH initiated reactions. A discussion of the technique of Nehr et al. and a comparison to the method used in this work will be included.

Material has been added in **lines 126-8**

'The instrument has some similarities to that presented by Nehr et al. (2011) where a conventional OH lifetime instrument was altered to allow for chemical conversion of $HO_2$ to OH and hence the sequential determination of OH and $HO_2$.'

and **616-629.**

'As mentioned in the introduction, the instrument has similarities to that presented by Nehr et al. (2011), where a FAGE system for sequential OH and $HO_2$ is coupled to a lifetime instrument and yields of $HO_2$ from OH initiated reactions are reported. Although the principles of HOx detection used in both systems is similar, there are some significant differences between the two instruments. Some differences relate to the reaction cell in which the kinetics takes place: 1 atm of air and 298 K for Nehr et al. and 0.5 – 5 atm of any gas and 298 – 800 K for this work. However, in principle, the Nehr et al. FAGE cell could be coupled to a different reaction cell to probe a wider range of conditions. A more substantial difference is the timescale of the chemistry taking place. Typical temporal profiles from Nehr et al. are of the order of a second compared to <10 ms in this work. The enhanced sensitivity of the Nehr et al. instrument means that radical-radical reactions should not interfere, but the technique may be subject to interferences from first order (or pseudo-first order) reactions including heterogeneous

processes. Detection of radicals in kinetics or yield experiments is difficult and studying reactions under a range of conditions is important to identify systematic errors, hence both instruments have a role to play.'

General remark to the figures: It would be easier to work with legends instead of descriptions in the captions.

**Response:**

Use of appropriate legends supported by detailed figure captions will indeed improve the overall readability of this work. See above comment.